# Impact of diet and host genetics on the murine intestinal mycobiome

Yask Gupta[1,2,5], Anna Lara Ernst [1,5], Artem Vorobyev[1,3,5], Foteini Beltsiou [1], Detlef Zillikens [3], Katja Bieber [1], Simone Sanna-Cherchi [2], Angela M. Christiano [4], Christian D. Sadik [3], Ralf J. Ludwig [1,3,5] ✉ & Tanya Sezin [3,4,5] ✉

The mammalian gut is home to a diverse microbial ecosystem, whose composition affects various physiological traits of the host. Next-generation sequencing-based metagenomic approaches demonstrated how the interplay of host genetics, bacteria, and environmental factors shape complex traits and clinical outcomes. However, the role of fungi in these complex interactions remains understudied. Here, using 228 males and 363 females from an advanced-intercross mouse line, we provide evidence that fungi are regulated by host genetics. In addition, we map quantitative trait loci associated with various fungal species to single genes in mice using whole genome sequencing and genotyping. Moreover, we show that diet and its' interaction with host genetics alter the composition of fungi in outbred mice, and identify fungal indicator species associated with different dietary regimes. Collectively, in this work, we uncover an association of the intestinal fungal community with host genetics and a regulatory role of diet in this ecological niche.

The mammalian gut is home to a diverse microbial ecosystem that harbors bacteria, archaea, fungi, protists, and viruses, which affect various physiological and pathophysiological mechanisms in the host[1-4]. Through recent advances in next generation sequencing-based meta-genomics, the critical role of host genetics in shaping of diverse microbial communities in mammalian gut, has been implicated in multiple developmental and pathogenic processes[5-7]. Considering environmental factors such as diet, stress, hygiene, among others, these emerging data provide important information for an even more detailed understanding of the complexity associated with alterations of these communities. Likewise, disruption of microbiota homeostasis (dysbiosis) has been established as a key driver of inflammatory diseases, such as inflammatory bowel disease (IBD) and systemic lupus erythematosus (SLE), and affects their clinical outcome by affecting the immune system of the host[8-11].

Given the high abundance of bacteria among all microbial gut communities and the well-established amplification methods of conserved hyper-variable bacterial gene regions, changes in other constituents of gut microbiota, such as fungi have not yet been studied in similar depth[12,13]. More recently, comparable to the 16S rRNA in bacteria, amplification and sequencing of the fungal ribosomal DNA ITS, specifically ITS1 and ITS2 has greatly facilitated our understanding of the biological role of fungi in the host. To date, whole metagenomic profiling and ITS1 and ITS2 sequencing have provided insights into the fungal kingdom within the gut ecosystem[14,15]. Recently, as a result of these advances, the important role of mycobiome, including the diversity and dynamics of fungi, in homeostasis as well as disease is being increasingly recognized[16-19].

Fungi belong to one of the largest and most diverse kingdoms of living organisms, and are associated with a variety of diseases[20-22].

[1]Lübeck Institute of Experimental Dermatology, University of Lübeck, Lübeck, Germany. [2]Division of Nephrology, Department of Medicine, Columbia University Irving Medical Center, New York, NY, USA. [3]Department of Dermatology, University of Lübeck, Lübeck, Germany. [4]Department of Dermatology, Columbia University Irving Medical Center, New York, NY, USA. [5]These authors contributed equally: Yask Gupta, Anna Lara Ernst, Artem Vorobyev, Ralf J. Ludwig, Tanya Sezin. ✉e-mail: ralf.ludwig@uksh.de; ts3391@cumc.columbia.edu

Previous research focused on gut bacterial communities and demonstrated that gut commensals are influenced by host genetics as well as external factors such as diet. Of note, caloric restriction led to higher diversities of gut microbiota, whereas the opposite was observed when the hosts were exposed to a Western diet[23–25]. However, the interplay of fungal communities, diet, and host genetics remains unknown.

We previously demonstrated the eminent impact of diet on the genetic susceptibility of NZM2410/J mice to develop SLE by reshaping the gut microbiome[26]. Caloric restriction in NZM2410/J mice led to a complete protection from clinical lupus manifestation, whereas accelerated disease was observed in NZM2410/J mice fed a Western diet. Thus, diet overrode genetic susceptibility and delayed SLE onset by reshaping intestinal bacterial and fungal communities before the onset of lupus. Consistent with this observation, disrupted interactions of bacteria and fungi and reduction of co-abundances between intra- and interkingdom microbial pairs were shown to be associated with disease pathogenesis in other murine models and in humans[27–32]. Thus, changes in microbial abundances can be linked to disease susceptibility, and together with the identification of host susceptibility genes, predict disease outcome[33–35].

To investigate the impact of host genetics and diet on the composition of the intestinal fungal ecosystems, in this study, we fed ~600 mice of an advanced-intercross mouse line (AIL) either a Western diet (WES), a control diet (CON), or a calorie-restricted diet (CAL) for a period of 5 months. At the end of the observation period, ITS2 and 16S rRNA sequencing was used to characterize the composition of the fungal and bacterial communities in the gut. Further, we used QTL mapping and a haplotype-sharing analysis to pinpoint possible associations of the gut microbiome with host genetics. We also investigated the impact of diet on the interaction between bacteria and fungi in the gut ecosystem. Overall, here we demonstrate that in the gut, fungal communities are associated with host genetics. Ultimately, we show that the composition and diversity of murine intestinal mycobiome is not exclusively regulated by host genetics, but by the joint interaction of diet and host genetics.

## Results

### Diet modulates fungal communities in the gut

To determine if diet modulates fungi composition in the gut, we randomly distributed 591 AIL mice at weaning to three different dietary groups and fed them three different diets: Western diet (WES), control diet (CON), or a calorie-restricted diet (CAL). At the age of six months, we collected cecum content from AIL mice and performed ITS2 sequencing (Fig. 1a and Supplementary Data 1). To characterize fungi composition across the three diets, we assigned reads from ITS2 sequencing to different taxonomical ranks from phylum to genus (Fig. 1b). At the phylum level, we identified *Ascomycota* and *Basidiomycota* as the most abundant phyla in the gut of AIL mice, which was inversely represented between CAL- and CON-diet fed mice, as opposed to WES-diet fed mice (Supplementary Data 2). Consistently, while *Ascomycota* was comparably abundant in CAL- and CON-diet fed mice and comprised up to 97% of all taxa in these mice, it was significantly decreased in WES-diet fed mice (91% of all taxa). In contrast, *Basidiomycota* made up 2% of all taxa in CAL- and CON-diet fed mice and was expanded in WES-diet fed mice (8%), suggesting that a high-fat diet promotes expansion of *Basidiomycota*. At the genus level, *Penicillium* was the most abundant genus (53.3%) found in the gut of all AIL mice, followed by *Aspergillus* at 8.4%, *unknown Ascomycota* at 7.8%, and *Candida* at 7.7%. Consistent with the decrease in the Ascomycota phylum, a significant decrease in several *Ascomycota* genera such as *Claviceps*, *Davidiella*, and *Alternaria*, and an expansion in the *Basidiomycota* genus, *Wallemia* was observed in WES-diet fed mice compared to CAL- and CON-diet fed mice (Fig. 1b). Detailed statistical analysis of relative abundances of all fungal taxa across the three dietary regimens is summarized in Supplementary Data 2.

To further examine gut fungal diversity and composition, we clustered the sequences to species level OTUs at a 97% threshold using the PIPITS pipeline[36] (Supplementary Data 3). No significant difference in fungal alpha diversity in terms of species richness and evenness (Chao1 index; Shannon index; Simpson index) was found among the three dietary groups (Fig. 1c). In contrast, variation of fungal communities (beta diversity) showed significant differences (Fig. 1d; $P_{CAL\ vs.\ CON} < 0.05$, $P_{CON\ vs.\ WES} < 0.01$; $P_{CAL\ vs.\ WES} < 0.01$). We used Linear discriminant analysis Effect Size (LEfSe) algorithm[37] (which determines features that explain differences between classes and assesses their relevance to the examined phenotype), to correlate fungi genera to the three dietary groups (Fig. 1e). The phylum *Basidiomycota* with the genera *Wallemia* and *Mycena*, and an unidentified genus from the order *Helotiales* were more abundant in WES- compared to CON- and CAL-diet fed mice. In the CAL-diet fed mice, the genera *Claviceps*, *Davidiella*, *Phoma,* and *Ascochyta*, and the family *Nectriaceae* all belonging to the *Ascomycota* phylum were more abundant compared to WES-diet fed mice.

Using indicator species analysis, in CAL-diet fed mice, *P. herbarum, A. nidulans*, and *N. paspali* were identified as indicator species (Fig. 1f and Supplementary Data 4). In contrast, in WES-diet fed mice, *W. sebi, P. decumbens, A. rubrum, F. culmorum, K. marxianus, P. chrysosporium, L. laevis,* and *N. gypsea* were identified as indicator species (Fig. 1f). Notably, several of the species identified in WES-diet fed mice were shown to be associated with allergic hypersensitization of airways and lung infections[38,39], chronic granulomatous disease[40,41], obesity[42,43], as well as chronic infections, and fatal mycoses in immunocompromised patients[44,45].

Taken together, our results show that diet modulates composition of intestinal fungi in the host by affecting their relative abundances in the gut. Further, WES diet promoted expansion of several *Basidiomycota* species, which were previously linked to inflammation and shown to be associated with development of innate and adaptive immune responses in mice[46,47].

### Composition of the gut bacteria and their alteration with diet

Since we identified that diet shifts fungi composition in the gut of AIL, we hypothesized that it may also modulate fungal bacterial interkingdom co-abundance correlations in the gut. To interrogate how diet shapes bacterial standing (DNA) and active communities (RNA) in the gut (Fig. 1a and Supplementary Data 1), we assigned rRNA gene copy (DNA) and transcript (RNA) levels from V1 to V2 region, respectively, to the 16S rRNA database using the RDP classifier. For standing communities, the phyla *Firmicutes* ($P = 1.68E{-}08$; CAL:64.7 ± 14%, CON:57 ± 11.7%, WES:53 ± 9.6%), *Bacteroidetes* ($P = 0.01$; CAL:28.1 ± 12.8%, CON:35.1 ± 11.4%, WES:25 ± 10.7%), and *Proteobacteria* ($P = 6.74E{-}30$; CAL:5 ± 5.2%, CON:5.2 ± 3.7%, WES:18.2 ± 7.5%) were most abundant in the gut of AIL mice, and were significantly different between the three groups of mice (Fig. 2a). Accordingly, WES-diet fed mice showed a significant decrease in the *Firmicutes* phylum and expansion of *Deferribacteres* and *Proteobacteria* phyla compared to CAL- and CON-diet fed mice. At the genus level, a significant decrease in *Lactobacillus* and *Mucispirillum* genera, as well as expansion of *Eisenbergiella, Bacteroides,* and *Anaeroplasma* genera was found in WES-diet fed mice compared to CAL-and CON-diet fed mice (Fig. 2a).

Detailed statistical analysis of the bacterial phyla and genera relative abundances across the three dietary regimens is summarized in Supplementary Data 5. Similarly, we found comparable trends in the overall distribution of major bacterial phyla across different diets in the active communities (Supplementary Fig. 1a; Supplementary Data 6).

To assess the diversity and the composition of gut bacteria at species-level resolution, we clustered the sequences at species level OTUs using the VSEARCH algorithm[48] (Supplementary Data 7 and Supplementary Data 8). We found significant changes in alpha

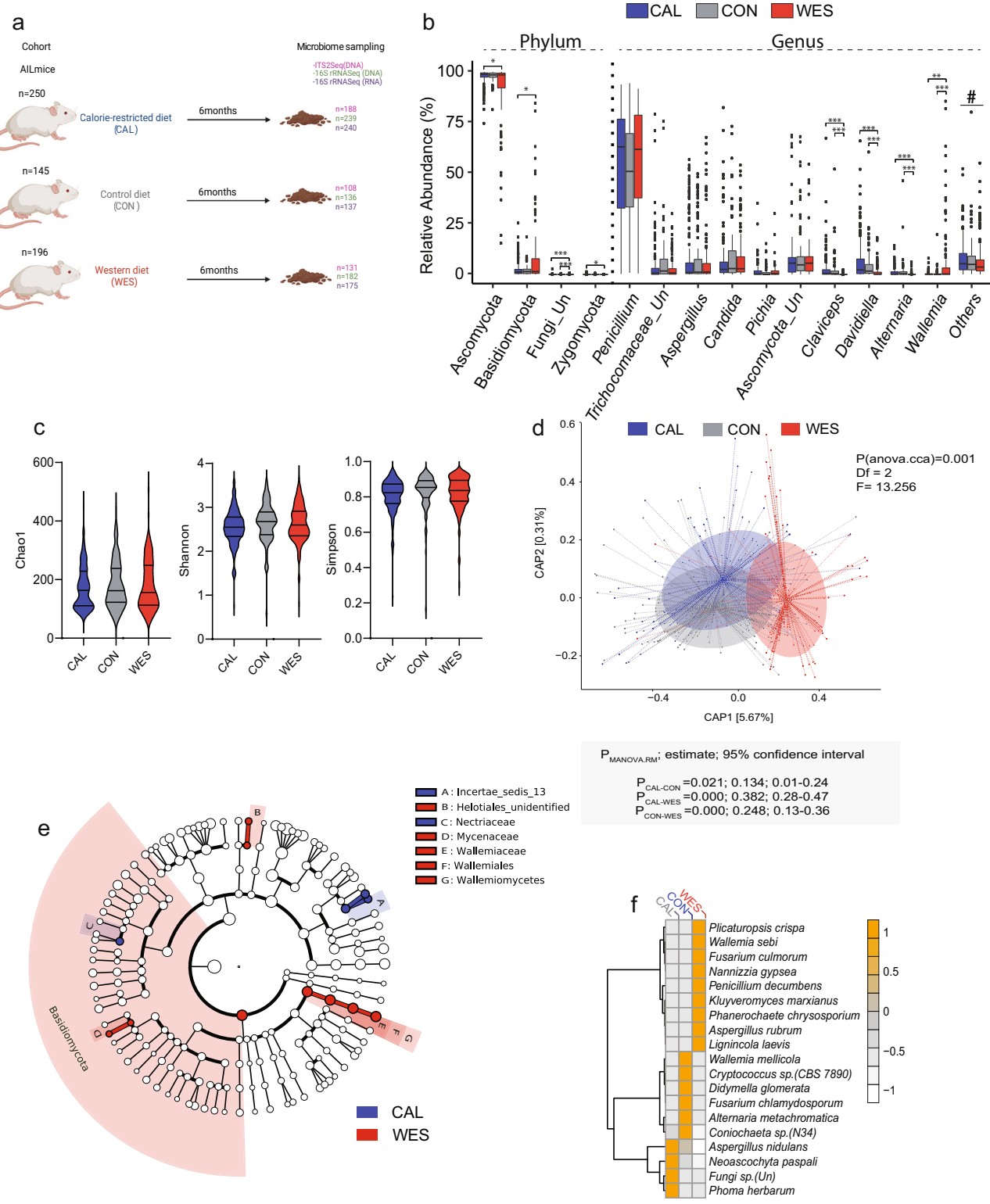

diversity (species richness and species evenness) as assessed by Chao1, Shannon, and Simpson indices among the three dietary groups in both the standing (Fig. 2b) and active communities (Supplementary Fig. 1b). In both type of communities, CAL-diet fed mice had a higher species diversity compared to CON-diet fed mice, whereas WES-diet fed mice showed the least diverse bacterial microbiome. Likewise, the beta-diversity was also significantly different between all the dietary groups for both standing ($P_{CAL\ vs.\ CON} < 0.001$, $P_{CON\ vs.\ WES} < 0.001$; $P_{CAL\ vs.\ WES} < 0.001$) and active communities $P_{CAL\ vs.\ CON} < 0.001$, $P_{CON\ vs.}$

$_{WES} < 0.001$; $P_{CAL\ vs.\ WES} < 0.001$) (Fig. 2c and Supplementary Fig. 1c). These results are consistent with previous reports on the influence of a high-fat diet on diminishing bacterial richness in the gut, and the effects of reduced microbial richness in promoting intestinal pro-inflammatory environment[49].

To further identify microbial taxa that were differentially repre-sented in CON-, CAL-, and WES-diet fed mice, we used the LEfSe algorithm[37]. We found the genera *Clostridiumsensustricto*, *Anaero-vorax*, and *Acetatifactor* were overrepresented in CAL-diet fed mice,

**Fig. 1 | Diet modulates composition of intestinal fungi in the host.**
**a** Experimental design (created using Biorender). Mice were fed calorie restricted diet (CAL; $n_{CAL}$ = 250; $n_{males}$ = 111; $n_{females}$ = 139), control diet (CON; $n_{CON}$ = 145; $n_{males}$ = 54; $n_{females}$ = 91), or western diet (WES; $n_{WES}$ = 196; $n_{males}$ = 63; $n_{females}$ = 133). We generated 427 ITS2 ($n_{males}$ = 167; $n_{females}$ = 260), 557 16S rRNA DNA ($n_{males}$ = 214; $n_{females}$ = 343), and 552 16S rRNA RNA ($n_{males}$ = 215; $n_{females}$ = 337) samples from cecum. **b** Box plots representing the relative abundance of fungal phyla and genera. CAL (blue; $n_{CAL}$ = 188; $n_{males}$ = 83; $n_{females}$ = 105), CON (gray; $n_{CON}$ = 108; $n_{males}$ = 39; $n_{females}$ = 69), and WES (red; $n_{WES}$ = 131; $n_{males}$ = 45; $n_{females}$ = 86) groups. The band in the box plot indicates the median, the box indicates the first and third QRs, and the whiskers indicate 1.5*IQR. (*Ascomycota*:$P_{CAL-WES}$ = 0.01; *Basidiomycota*: $P_{CAL-WES}$ = 0.01; *Fungi_Un*: $P_{CAL-WES}$ = 1.24 × 10$^{-5}$, $P_{CON-WES}$ = 0.0009; *Zygomycota*: $P_{CAL-WES}$ = 0.04; *Claviceps*: $P_{CAL-WES}$ = 3.79 × 10$^{-30}$, $P_{CON-WES}$ = 2.68 × 10$^{-18}$; *Davidiella*: $P_{CAL-WES}$ = 1.35 × 10$^{-11}$, $P_{CON-WES}$ = 6.59 × 10$^{-7}$; *Alternaria*: $P_{CAL-WES}$ = 1.55 × 10$^{-13}$, $P_{CON-WES}$ = 1.31 × 10$^{-10}$; *Wallemia*: $P_{CAL-WES}$ = 0.001, $P_{CON-WES}$ = 0.0001). #Supplementary Data 2 shows statistical analysis of all taxa including low abundant taxa ("others") across different diets. **c** Violin plots depicting alpha diversity indices of mycobiota composition in mice. The lines in the violin plot from bottom to top indicate 1st QR, median, and 3rd QR. The tips of the violin plot represent minima and maxima, and the width of the violin plot shows the frequency distribution of the data. **d** Beta diversity of intestinal fungi composition in mice displayed by canonical analysis of principal coordinates (capscale) plot of the BrayCurtis distances. **e** Differentially abundant mycobiota taxa identified by LEfSe algorithm in CAL (blue) and WES (red) mice. The root denotes the fungal domain and size of each node corresponds to the relative abundance of the taxon. **f** Heatmap showing fungal indicator species and mean scaled counts for every species within each diet. Statistical significance in panel **b** was determined using Kruskal–Wallis test followed by two-sided Mann–Whitney U test adjusted by FDR correction. In panel **c** statistical significance was determined using one-way ANOVA on residuals after sex and generation adjustment followed by Tukey's multiple comparisons test. *Padj < 0.05, **Padj< 0.01, ***Padj< 0.01. Data in panel **d** was analyzed using "anova.cca" function (999 permutations) followed by "MANOVA.RM" for post hoc analysis. Source data for **b**–**f** are provided as a Source Data file.

and the genus *Bacteroides* was overrepresented in WES-diet fed mice, in both the active and standing communities (Fig. 2d and Supplementary Fig. 1d). These findings are consistent with the previously reported increase in the abundance of *Bacteroides* in WES-diet fed mice, and the observed shift towards metabolism of simple sugars and lipid digestion[50,51]. In contrast, the genus *Acetatifactor* (which was overrepresented in CAL-diet fed mice) was previously reported to be involved in production of acetate and butyrate in the gut, which suppress inflammatory response and regulate insulin sensitivity[52,53]. Additionally, *Acetatifactor* has also been reported to regulate the development of obesity by improving the intestinal absorption of dietary fats[54,55].

Using indicator species analysis, we identified species from *Clostridium IV* (C. hylemonae, C. populeti, C. xylanolyticum and C. tertium), *Alistipes* (A. onderdonkii and A. putredinis), *Lactobacillus* (L. hominis and L. reuteri), A. agile, B. coccoides, C. eutactus, D. longicatena, and S. saccharolytica as indicator species in the gut of CAL-diet fed mice. In contrast, B. pectinophilus, B. wadsworthia, E. oxidoreducens, F. orotica, F. pleomorphus, L. bovis, O. valericigenes, P. capillosus, and R. hominis, were identified as indicator species in the gut of WES-diet fed mice in both the standing and active communities (Fig. 2e, Supplementary Fig. 2a, b, Supplementary Fig. 1e, Supplementary Data 9, and Supplementary Data 10).

Consistent with previous reports, we demonstrated that diet shifts the composition of standing and active bacterial communities in the gut, affecting both species richness and evenness, as well as their relative abundances. Furthermore, we identified overrepresentation of the genus *Bacteroides* and the genus *Acetatifactor*, in the gut of WES-diet and CAL-diet fed mice, respectively.

## Inter-domain correlations of fungi and bacteria in the gut ecosystem

We and others previously identified inter-domain co-abundances correlations between microbial and fungi communities in the gut to be associated with inflammatory responses in the host[26,28,56]. To investigate associations between fungal and bacterial communities and to characterize how different kingdoms interact within the murine gut ecosystem, we correlated fungal genera abundances with bacterial standing ($n_{DNA}$ = 419) and active ($n_{RNA}$ = 420) genera across the three different diets using FastSpar algorithm[57]. (Fig. 3a, Supplementary Data 11, and Supplementary Data 12). We observed 320 significant correlations based on 1000 bootstrap permutations ($P < 0.05$) between fungi and standing bacterial communities (Fig. 3a and Supplementary Data 11), and 323 significant correlations between fungi and active bacterial communities (Fig. 3a and Supplementary Data 12). Out of 323 correlations, 174 (37%) correlations were common between bacteria and fungi, and more than 90% of all correlations were negative (DNA = 90.6%; RNA = 90.1%), suggesting that bacteria and fungi

negatively regulate each other co-abundances in the gut. These findings highlight to a competitive relationship between fungi and bacteria in the gut[58] and are consistent with the observation that prolonged use of systemic antibiotics promotes invasive fungal infection in the host and fungal overgrowth in the gut[59,60].

The strongest negative correlation with fungal genera was observed between *Intestinimonas* and *Claviceps* for standing bacterial communities and between *Butyricicoccus* and *Wallemia* for active bacterial communities. In contrast, the strongest positive correlation with fungal genera was observed between *Lactobacillus* and the genus *Claviceps* for both standing and active communities. In addition, we observed a significant ($P < 0.05$) negative correlation between *Lactobacillus* and *Candida* in our study, further supporting the previously reported antagonistic effects of *Lactobacillus* on the growth of *C. albicans* in the gut via the production of lactic acid and hydrogen peroxide[61,62]. The genus *Claviceps* showed the highest number of negative associations ($n_{cor}$=30) with standing bacterial communities (Fig. 3a), while the genus *Candida* was mostly associated ($n_{cor}$ = 24) with active communities (Supplementary Fig. 3). Consistently, alkaloids of *Claviceps* (also known as ergot) and their derivatives were shown to display antimicrobial properties[63]

Further, we investigated conserved correlations (common to both active and standing communities; $P_{adj} < 0.05$) between bacteria and fungi (Fig. 3b). We observed nine putative key drivers (degree ≥ 10) within the bacteria and fungus kingdoms in the gut. These include five fungi genera (*Claviceps, Candida, Alternaria, Davidella* and *Wallemia*) and four bacterial genera (*Lactobacillus, Clostridium IV, Butyricicoccus* and *Pseudoflavonifractor*). Further, altered abundances of *Candida, Alternaria, Wallemia*, and *Lactobacillus* were reported in the gut of the dextran sulfate sodium-induced colitis mouse model[64]. Additionally, dysbiosis characterized by underrepresentation of the butyrate-producing *Clostridium* cluster *IV* genus *Butyricicoccus* was identified in the gut of IBD patients[65].

Taken together, our findings suggest that both bacteria and fungi play central role in maintaining the homeostasis in the gut ecosystem by inversely regulating each other composition in the gut.

## The intestinal fungal community shows a strong genetic association

Having identified that diet modulates intestinal fungi composition in AIL mice, we next addressed if host genetics and its interaction with diet is associated with differences in the intestinal mycobiome. To interrogate the relationship of host genetics with the fungal community in the gut, we determined changes in intestinal fungi composition in association with variations in host genetics (additive model), host genetics interaction with diet (IntDiet model) or sex (IntSex model) in AIL mice. We used 591 mice from the AIL population across three generations (18th–20th) and across WES, CON, or CAL, diets to

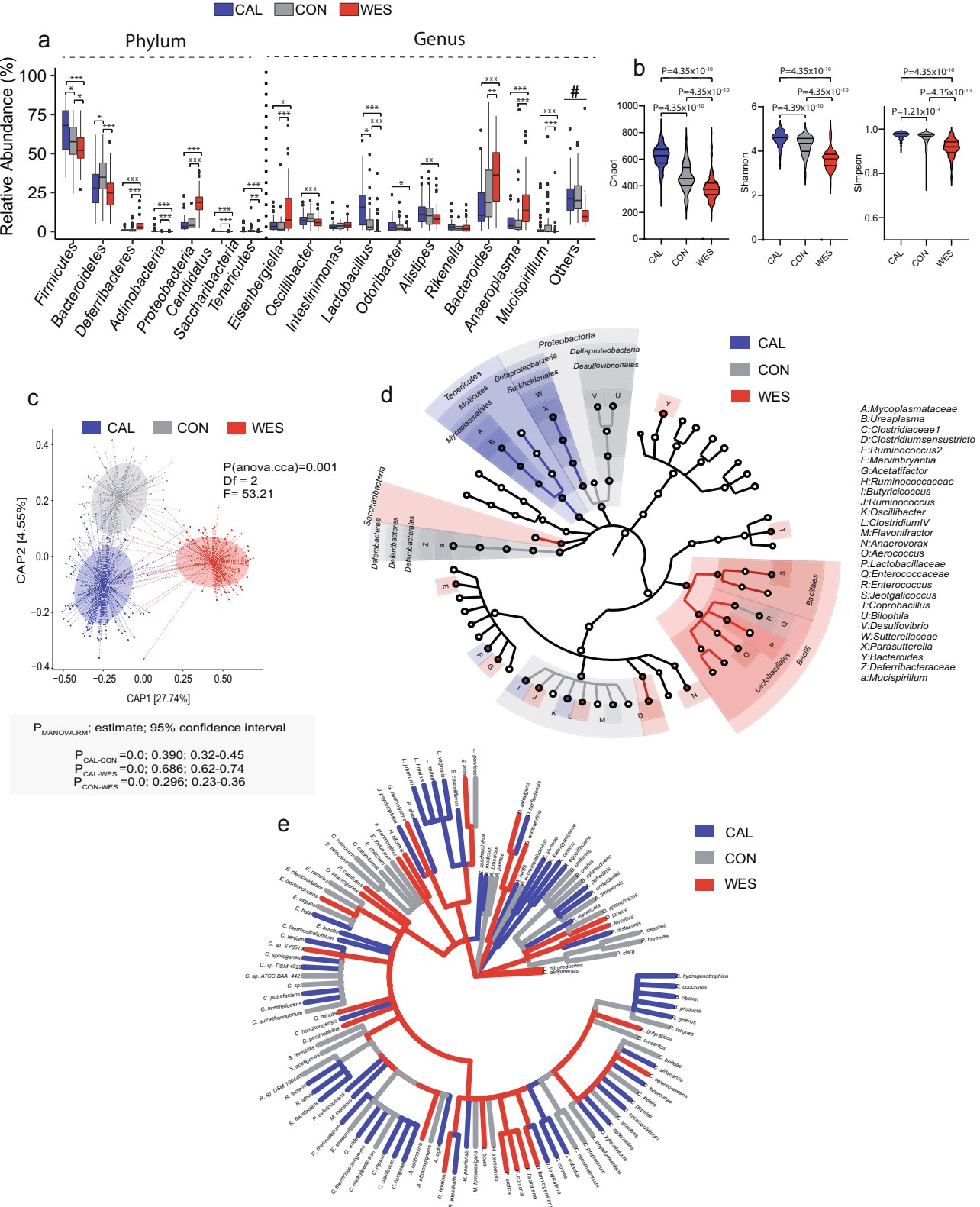

determine an estimate of QTL map location compared to conventional mapping populations (Figs. 1a and 4). In AIL mice, the average size of the QTL that were identified at genome-wide-significance level ($\alpha$gw < 0.05) using 1000 permutations for each taxon and for each model (Additive, IntDiet, and IntSex) was $0.91 \pm$ (SEM) 0.12 Mb (Table 1).

In total, we mapped 51 QTL (genome-wide $p$-value < 0.05) that influence the abundance of 42 intestinal fungal taxonomic lineages in

AIL mice. We identified 28 QTL for the Add model (A), 17 QTL for the IntDiet (D) model, and 6 QTL for IntSex (S) model (Table 1). Consistent with previous findings, the highest percentage of phenotypic variation in fungi composition that were associated with host genetics was explained by cage (mean: 24.7%)[66,67]. Further, host genetics explained an average 9.1% of the phenotypic variation in AIL mice, whereas diet accounted for 1.0% of the phenotypic variation. Intrinsic factors such as generation and sex explained 4.7 and 0.05% of the phenotypic

**Fig. 2 | Diet modulates composition of bacterial standing communities in the host. a** Box plots representing relative abundance of bacterial phyla and genera. The band in the box plot indicates the median, the box indicates the first and third QRs, and the whiskers indicate 1.5*IQR. Calorie-restricted (CAL; blue; $n_{CAL}$ = 239; $n_{males}$ = 106; $n_{females}$ = 133), control (CON; gray; $n_{CON}$ = 136; $n_{males}$ = 51; $n_{females}$ = 85) or WES (Wes; red; $n_{CAL}$ = 182; $n_{males}$ = 57; $n_{females}$ = 125) groups. (*Firmicutes*: $P_{CAL-CON}$ = 0.01, $P_{CAL-WES}$ = 2.97 × 10⁻⁸, $P_{CON-WES}$ = 0.04; *Bacteroidetes*: $P_{CAL-CON}$ = 0.01, $P_{CON-WES}$ = 2.92 × 10⁻⁶; *Deferribacteres*: $P_{CAL-WES}$ = 6.62 × 10⁻¹³, $P_{CON-WES}$ = 9.91 × 10⁻¹⁵; *Actinobacteria*: $P_{CAL-WES}$ = 5.05 × 10⁻¹⁸, $P_{CON-WES}$ = 8.93 × 10⁻¹²; *Proteobacteria*: $P_{CAL-WES}$ = 1.43 × 10⁻²⁶, $P_{CON-WES}$ = 1.79 × 10⁻²¹; *Candidatus Saccharibacteria*: $P_{CAL-WES}$ = 2.51 × 10⁻¹⁹, $P_{CON-WES}$ = 1.77 × 10⁻¹³; *Tenericutes*: $P_{CAL-WES}$ = 0.0008, $P_{CON-WES}$ = 0.003; *Eisenbergiella*: $P_{CAL-WES}$ = 0.01, $P_{CON-WES}$ = 6.39 × 10⁻⁶; *Oscillibacter*: $P_{CON-WES}$ = 0.0005; *Lactobacillus*: $P_{CAL-CON}$ = 0.01, $P_{CAL-WES}$ = 7.24 × 10⁻²⁶, $P_{CON-WES}$ = 2.15 × 10⁻¹³; *Odoribacter*: $P_{CAL-WES}$ = 0.01; *Alistipes*: $P_{CAL-WES}$ = 0.007; *Bacteroides*: $P_{CAL-WES}$ = 1.39 × 10⁻⁸, $P_{CON-WES}$ = 0.002; *Anaeroplasma*: $P_{CAL-WES}$ = 9.41 × 10⁻⁹, $P_{CON-WES}$ = 6.39 × 10⁻⁶; *Mucispirillum*: $P_{CAL-WES}$ = 1.47 × 10⁻⁸, $P_{CON-WES}$ = 2.22 × 10⁻¹¹).⁑Supplementary Data 5 shows statistical analysis of all taxa including low abundant taxa ("others") across different diets. **b** Violin plots depicting alpha diversity indices of bacterial standing communities in mice. The lines in the violin plot from bottom to top indicate 1st QR, median, and 3rd QR. The tips of the violin plot represent minima and maxima, and the width of the violin plot shows the frequency distribution of the data. **c** Capscale plot of the BrayCurtis distance depicting beta diversity of bacterial standing communities in mice. **d** Differentially abundant microbial taxa of bacterial standing communities identified by the LEfSe algorithm in CAL- (blue), CON (gray), and WES- (red) mice. The root represents the fungal domain and the size of each node corresponds to the relative abundance of the taxon. **e** Cladogram depicting bacterial indicator species among standing communities for each diet. Statistical significance in panel **a** was determined using Kruskal–Wallis test followed by two-sided Mann–Whitney U test adjusted by FDR correction. In panel **b** statistical significance was determined using one-way ANOVA on residuals after sex and generation adjustment followed by Tukey's multiple comparisons test.*Padj < 0.05, **Padj < 0.01, ***Padj < 0.001. Data in panel **c** was analyzed using "anova.cca" function (999 permutations) followed by "MANOVA.RM" for post hoc analysis. Source data for **a**–**d** are provided as a Source Data file.

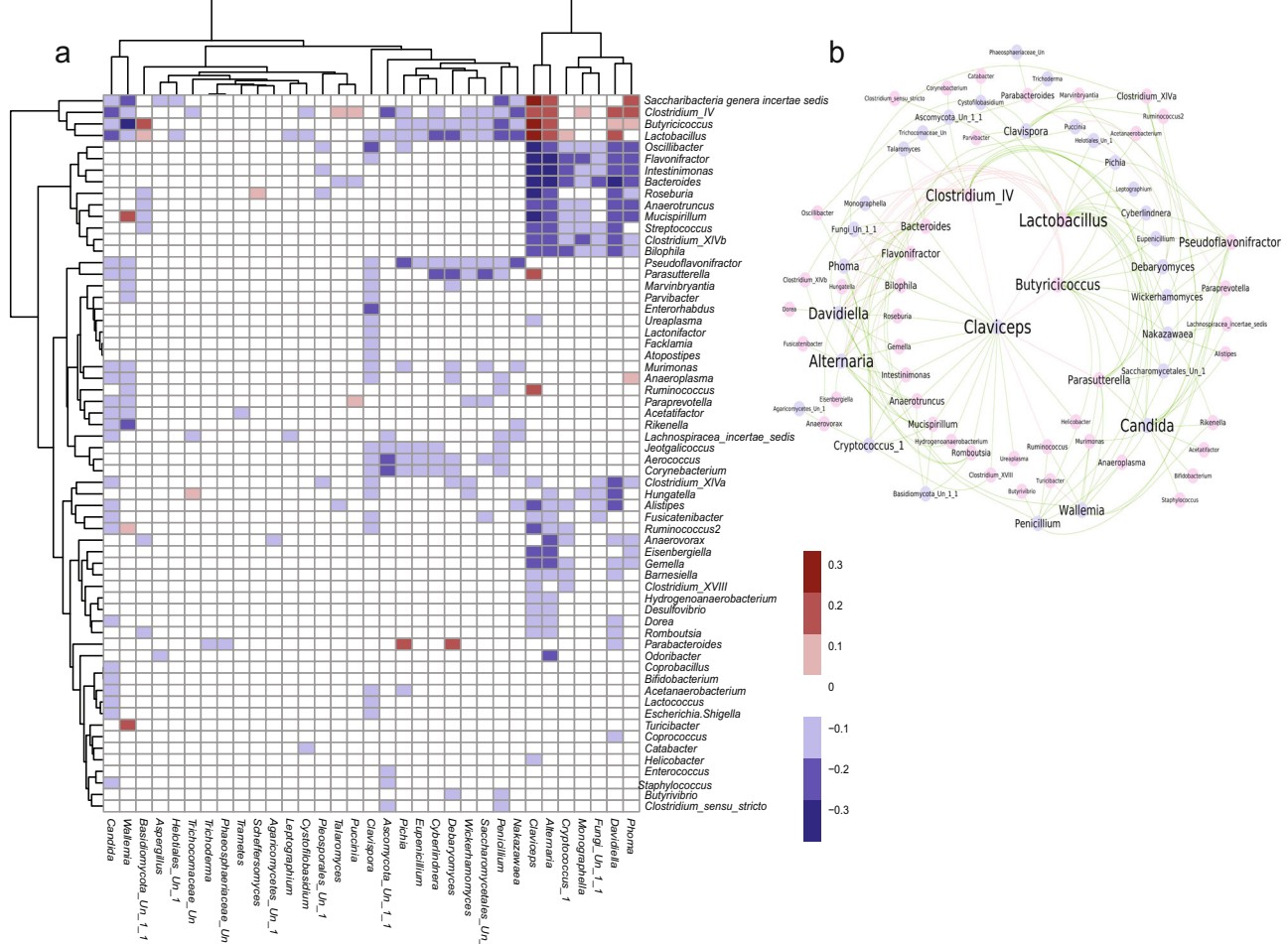

**Fig. 3 | Interaction network between microbial standing communities and fungi in the gut of AIL mice. a** Heatmap demonstrating significant (Padj < 0.05) correlations between fungal genera (columns) abundances and bacterial genera (rows) abundances (standing communities). The color codes of the cells indicate either positive (purple) or negative (orange) correlations among the species (Padj < 0.05); *n* = 419 samples were used for the correlation analysis. **b** Interaction network demonstrating conserved correlations (interaction that is present for both standing and active communities; Padj < 0.05) between bacterial standing communities (pink nodes) and fungal (blue nodes) taxa. The size of the font indicates the key driving taxa, which are characterized as taxa with the highest number of associations with taxa from the other domain. Positive correlations are denoted in red and negative correlations are denoted in green. Data in **a**, **b** were calculated using FastSpar implementation of the FastCC algorithm (*n* = 999 permutations). P-value were adjusted using Benjamini–Hochberg correction. Source data are provided as a Source Data file.

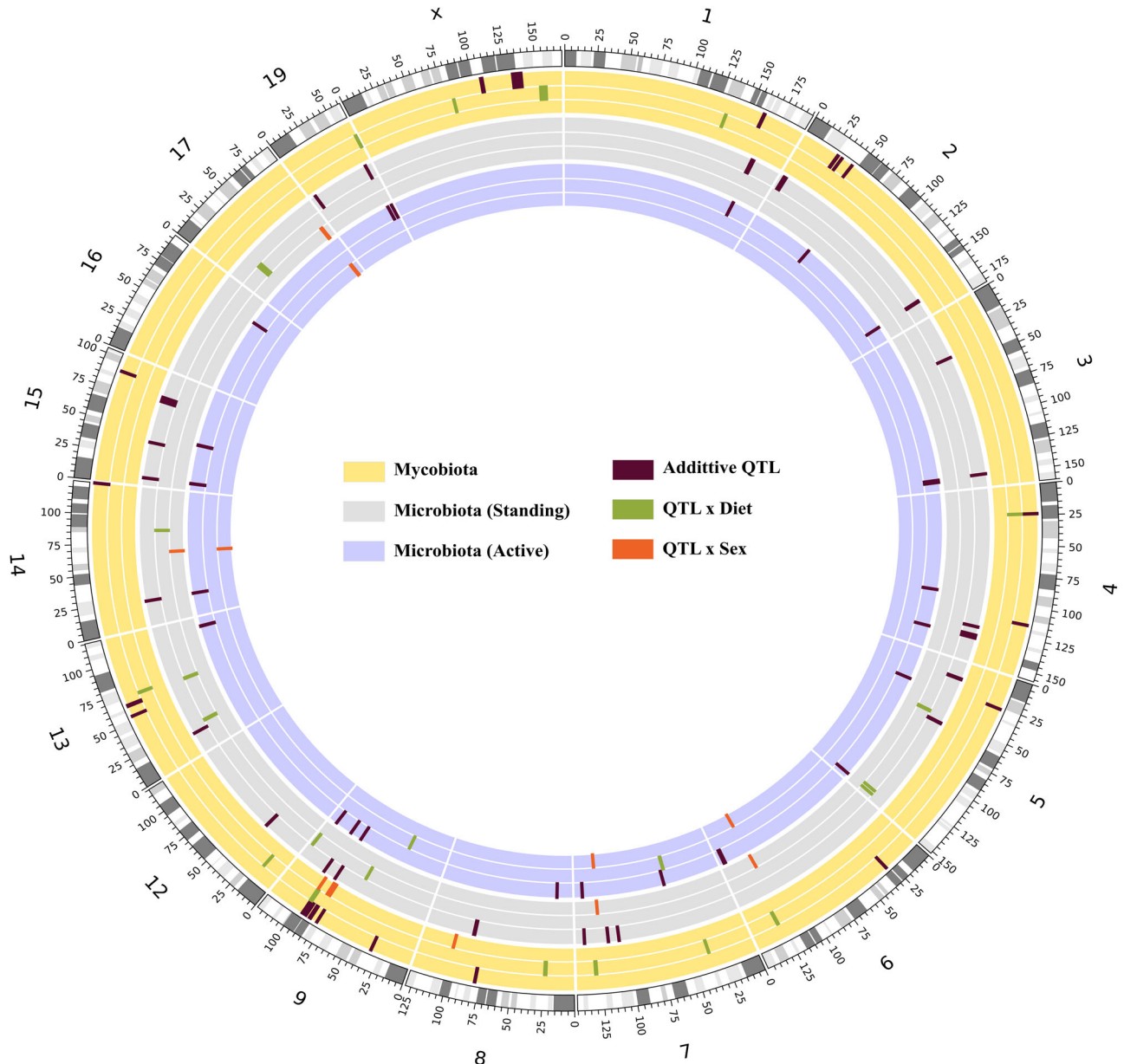

**Fig. 4 | Gut fungal and bacterial QTL in AIL mice.** Circos plot showing QTL (αgw < 0.05) in AIL mice associated with bacteria (standing and active community) and fungi as traits. The outer most circle in the circos plot describes chromosomes with cytogenic bands in the C57BL/6J mouse genome (reference mouse assembly mm10). Every circle within each chromosome is color coded for fungi (yellow), bacterial standing communities (gray) and bacterial active communities (purple) variation, respectively (Table 1). Note, only chromosomes in which QTL were detected are shown. QTL, quantitative trait loci.

describing the three models i.e., additive (Add), interaction with diet (IntDiet), and interaction with sex (IntSex) from outwards to inwards. The QTL, marked as rectangles, with brown color represent the additive model, while the green color and the red color represent diet- and sex-interacting QTL, respectively. Note, only chromosomes in which QTL were detected are shown. QTL, quantitative trait loci.

variation, respectively (Table 1). Taken together, these findings suggest that host genetics only, as well as its interaction with diet modulate the composition of fungi in the gut in mice.

**Mapping of fungal QTL**

Since we identified that host genetics alone, or in combination with diet or sex regulates the composition of fungi in the gut, we next explored the identified QTL and the associated candidate genes, for their potential function in modulating fungi composition at different taxonomical levels. At the phylum level and consistent with the effects of diet on *Basidiomycota* composition in the gut[68], we identified two IntDiet QTL on chromosome (Chr) 7 (0.16 Mb; LOD = 10.3) and Chr 11 (0.98 Mb; LOD = 9.0) containing the genes *Gm23292* and *Tmc8*, respectively. Rare variants in the *TMC8* gene, which encodes for a

transmembrane channel-like protein, were previously discovered to be associated with type 2 diabetes (T2D) and obesity in Russian and Japanese patient cohorts[69,70]. Additionally, variants in the *TMC8* locus were associated with the levels of glycated hemoglobin (HbA1c), a commonly used test used in clinical settings to assess glycemic control and an established biomarker and predictor for T2D and cardiovascular disease[71].

At the class level, we identified IntSex QTL from an unknown Basidiomycota on Chr 9 (0.61 Mb; LOD = 8.95), one Add QTL for the class *Pucciniomycetes* (Chr 9; 0.02 Mb; LOD = 8.4*)*, and IntDiet QTL for the composition of *Zygomycota* (Chr7; 1.1 Mb; LOD = 11.1) containing the *Wdr11* gene. *Zygomycota* is a class of opportunistic fungi, which was previously reported to cause infections in immunocompromised hosts with severe underlying metabolic disorders[72,73]. Consistently, the gene

# Table 1 | Fungal QTL and the identified candidate genes

| RDP (UNITE)/OTU ID | Taxonomic name | Taxonomic rank | Chromosome | SNP | LOD | START (BP) | END (BP) | DIFF (Mb) | %h² (LOD) | %h² (Sex) | %h² (Gen) | %h² (Diet) | %h² (Cage) | ρ (Sex) | ρ (Gen) | ρ (Diet) | Model | Candidate genes |
|---|---|---|---|---|---|---|---|---|---|---|---|---|---|---|---|---|---|---|
| 2 | Basidiomycota | Phylum | 7 | UNC12630138 | 10.35 | 36217417 | 36377617 | 0.1602 | 10.8 | 0.2 | 2 | 0.79 | 35.85 | 0.0672 | -0.0234 | 0.0718 | IntDiet | Gm23292 |
| 2 | Basidiomycota | Phylum | 11 | UNC20492040 | 9.07 | 117545132 | 118530429 | 0.985297 | 9.53 | 0.2 | 2 | 0.79 | 35.85 | 0.0672 | -0.0234 | 0.0718 | IntDiet | Tmc8,6030468B19Rik,Gm20708,Tha1,Dnah17,Usp36,Tmrc6c,Tmrc6a,Tmc6,Tkt,Afmid,Bircs,Tmem235,Socs3,Pgs1,Rbfox3,Syngr2,Cyth1,Timp2,Cap295nl,Lgals3bp,Cant1,Clqtnf1,Engase |
| 1616 | Zygomycota/Incertae_sedis_10 | Class | 7 | UNC13835983 | 11.19 | 129034744 | 130149978 | 1.115234 | 11.63 | 0.1 | 2.12 | 0.53 | 1.21 | 0.1847 | -0.0781 | 0.034 | IntDiet | Pipp4 Wdr11 |
| 112 | Polyporales | Order | 6 | UNC12042546 | 10.37 | 126848048 | 127850668 | 1.00262 | 10.82 | 0.13 | 0.31 | 0.93 | 15.06 | 0.0446 | -0.0563 | 0.0313 | IntDiet | Akap3,Rad51ap1,Cracr2a,Ndufa9,Dyrk4,D6wsu163e,Fgf6,Fgf23,Tigar,Parp11,Prmt8 |
| 40 | Russulales | Order | 9 | UNC16882054 | 11.7 | 93022854 | 94468212 | 1.445358 | 12.12 | 0.08 | 4.08 | 0.48 | 9.95 | 0.0365 | -0.0689 | 0.0062 | IntDiet | Gm24200 |
| 121 | Cantharellales | Order | 18 | JAX00081898 | 9.39 | 31199252 | 32034289 | 0.835037 | 9.85 | <0.01 | 0.75 | 0.23 | 5.22 | 0.123 | -0.0493 | -0.0058 | IntDiet | Myo7b,Rit2,Syt4,Slc25a46,Sap130,Ammecr1,Polr2d,Wdr33,Sft2d3,Lim52,Gpr17 |
| 2156 | Hypocreales_unidentified | Family | 13 | UNC22815472 | 9.97 | 70348004 | 71076189 | 0.728185 | 10.43 | 0.01 | 3.32 | 0.1 | 13.54 | 0.1903 | -0.0743 | 0.0194 | IntDiet | Ice1,Adamts16 |
| 2314 | Corticiaceae | Family | 18 | UNC28777523 | 13.46 | 14637698 | 15762396 | 1.124698 | 13.81 | 0.07 | 0.94 | 0.45 | 5.92 | 0.1452 | -0.0942 | -0.1013 | IntDiet | Psma8,Ss18, Taf4b, Kctd1,Aqp4,Chst9 |
| 2157 | Hypocreales_unidentified_1 | Genus | 13 | UNC22815472 | 8.86 | 70693059 | 70712481 | 0.019422 | 9.32 | <0.01 | 3.43 | 0.1 | 13.64 | 0.1893 | -0.0736 | 0.0203 | IntDiet | Gm36607 |
| 5603 | Vuilleminia | Genus | 18 | UNC28777036 | 9.63 | 14637698 | 16134939 | 1.497241 | 10.09 | 0.13 | 1.23 | 0.06 | 4.33 | 0.1743 | -0.1016 | -0.0759 | IntDiet | Psma8,Ss18, Taf4b, Kctd1,Aqp4,Chst9 |
| 1758 | Geosmithia | Genus | X | UNC31499614 | 10.76 | 153533292 | 158259643 | 4.726351 | 11.2 | 0.02 | 1.15 | 0.08 | 13.96 | 0.2821 | -0.2599 | 0.0715 | IntDiet | Cypt3,Samt4,Magee6,Kctd12b, Nbdy, Spin2c,Samt1,Samt2,Cldn34b1, Cldn34b2, Magea8,Magea1, Magea2, Magea5, Sat1, Acon9,Prdx4, Pichdl,Phex, Sms, Kihl34, Cnksr2,Yy2,Mbtps2,Smpx |
| OTU2352 | Penicillium citoenigrum | Species | 1 | CEAJAX0009745 | 11.7 | 133221151 | 133859717 | 0.637566 | 11.86 | 0.01 | 7.31 | 0.23 | 19.14 | 0.0075 | -0.1375 | -0.0268 | IntDiet | Kiss1,Snrpe,Zc3h11a,Atp2b4,Plekhle6, Goltra,Gm28040,Etnk2,Sox13, Gm38394,Zbed6,Lax1 |
| OTU29 | Malassezia restricta | Species | 8 | UNC080619407 | 8.77 | 22671231 | 23905921 | 1.23469 | 9.03 | <0.01 | 1.28 | 0.33 | 16.16 | -0.0619 | -0.0125 | <0.001 | IntDiet | Plat,Kaf6a,Ank1,Ikbkb |
| OTU2243 | Malassezia restricta | Species | 18 | UNC29116030 | 9 | 41708067 | 41733945 | 0.025878 | 9.25 | <0.01 | 12.97 | 0.54 | 29.59 | -0.0099 | -0.3142 | 0.0578 | IntDiet | Nearest gene AC154172.2,Gm35835, Gm48784, Gm35940 |
| OTU2213 | Penicillium spathulatum | Species | 18 | UNC29082687 | 9.14 | 39047223 | 39715173 | 0.66795 | 9.38 | <0.01 | 13.05 | 0.44 | 31.97 | -0.0385 | -0.3344 | 0.0514 | IntDiet | Nr3c1,Arhgap26 |
| OTU298 | Aspergillus nidulans | Species | 19 | UNC30601828 | 8.87 | 60383903 | 61103683 | 0.71978 | 9.13 | 0.04 | 3.97 | 3.84 | 34.48 | 0.0137 | 0.1997 | -0.2017 | IntDiet | Fam45a,Cacult, Prthr, Nanos1, Eif3a, Prdx3, Sfxn4, Grk5 |
| OTU941 | Aspergillus glabripes | Species | X | UNC200105170 | 9.77 | 82219367 | 82464595 | 0.245228 | 10 | 0.04 | 3.97 | 0.01 | 86.09 | -0.033 | 0.2072 | -0.0592 | IntDiet | Gm24706 |

The table shows all mapped QTL derived from fungal DNA sequences across different models Add (host genetics-mycobiome interactions only), IntDiet (host genetics-diet interactions only), and IntSex (host genetics-sex interactions only) with taxonomical assignment of fungi from phylum to species level on chromosomal loci. Percentages of phenotypic variation for each QTL explained by sex, genetics, generation, diet, and cage are indicated as %h² LOD, %h² Sex, %h² Generation (Gen), %h² Diet, and %h² Cage, respectively. Also, spearman's correlation coefficients are displayed as ρ (Sex), ρ (Diet), and ρ (Gen).

The direction of the regression coefficient were obtained by converting categorical variable to quantitative variable.

Sex as males =1 and females = 0; Diet as calorie-reduced = 0, control =1 and western = 2; Generation = 18, 19, and 20.

*WDR11* was previously identified in a meta-analysis of genome-wide association study for adiponectin levels in East Asian individuals, which positively correlate with increased risk for metabolic and cardiovascular disorders[74].

At the order level, we identified *Corticiales* (IntSex; 0.55 Mb; LOD = 9.16), and its family *Corticiaceae* and the genus *Vuilleminia* on Chr 18. Notably, a QTL for the family *Corticiaceae* and its genus *Vuilleminia* was identified using all the three models, i.e., Add, IntSex, and IntDiet with interweaving loci containing the gene *Taf4b*. TATA-Box Binding Protein Associated Factor 4b (*Tafb*) and its paralog gene *Taf4* form the TFIID protein complex during gene transcription by RNA polymerase II. *Taf4b* maintains TFIID integrity in the absence of *Taf4*, which was previously shown to regulate insulin signaling pathway in pancreatic beta cells in mice[75].

At the species level, we identified 19 QTL (13 Add QTL; 6 IntDiet QTL) corresponding to 15 species, out of which ~95% belonged to the Ascomycota phylum and one species (*Malassezia restricta*) belonged to the *Basidiomycota* phylum. The *Ascomycota* species include predominantly members of the *Eurotiales* (10/15 species) and *Saccharomycetales* (3/15 species) orders. For example, two QTL for the *Eurotiales* species, *Aspergillus nidulans* were identified on Chr 11 (Add; 0.54 Mb; LOD = 6.72) and Chr 19 (IntDiet; 0.71 Mb; LOD = 8.87). The IntDiet QTL for *A. nidulans* contains the candidate gene *Grk5*, which was previously associated with metabolic disorders[76,77]. Increased expression of *GRK5* has been reported in several pathologies including cardiac hypertrophy and heart failure, hypertension, cancer, obesity, and T2D. Additionally, *Grk5*−/− mice were shown to exhibit higher glucose, insulin, and triglyceride levels, and were more resistant to insulin. Several case reports, demonstrating *A. nidulans* infection in T2D patients were also recently reported[78,79]. Other species, including the species *Yamadazyma Mexicana* from the *Saccharomycetales* order was mapped to Chr 11 (Add; 3.9 Mb; LOD = 6.98) containing the candidate genes *Nos2* and *Vtn*. Polymorphisms in *Nos2* gene encoding for the inducible nitric oxide synthase 2 (NOS2) were previously reported in IBD patients[80]. Accordingly, higher activity of NOS2, which contributes to the microbicidal activity of macrophages in the host, were found to contribute to onset of IBD in children below 10 years of age[81]. Further, increased abundance of *Saccharomycetales* was previously demonstrated in the ileum of patients with ulcerative colitis and Crohn's disease[82,83]. In addition to *Nos2*, the candidate gene *Vtn* that was identified in the QTL for *Y. Mexicana* and encoding for the Vitronectin protein is known to interact with β-glucans of the fungal cell wall and induce the release of TNF-α by macrophages[84]. At last, we identified IntDiet QTL (Chr 8; 1.23 Mb; LOD = 8.77) for the Basidiomycota species *M. restricta*, containing the *Ikbkb* gene (also known as the inhibitor of nuclear factor kappa-B kinase subunit beta or IKBKB). IKBKB phosphorylates IkB molecule, which interacts with the NF-kB transcription factors (RelA and p50) and renders them inactive. Once phosphorylated, the IkB molecule is degraded followed by the release of NF-kB transcription factors and activation of the NF-kB pathway in the nuclei. Thus, various signaling pathways that activate the NF-kB transcription factors converge at the level of IKKb. Interestingly, *M. restricta* was recently shown to promote the production of NF-kB−mediated cytokines (TNF-α and IL-6) in myeloid phagocytes in the gut of patients with Crohn's disease in a CARD9-dependent manner[85]. CARD9 serves as an adaptor for c-type lectin receptors, which recognize fungi and activates the NF-kB signaling pathway. Consequently, loss-of-function mutations in CARD9 were recently shown to impair anti-fungal IgG responses, thereby regulating fungi distribution in the gut[86].

Diet dependent QTL (IntDiet) were observed for five species *Penicillium citreonigrum*, *M. restricta*, *Penicillium spathulatum*, *A. nidulans* and *Aspergillus glabripes*. Deconvolving AIL mice to the founder strains (see "Methods" section) showed higher abundance of *A. nidulans*, *M. restricta* and *P. spathulatum* in AIL mice containing MRL/MpJ allele under CAL and CON diets in comparison to WES diet

(Supplementary Fig. 4a–d). Same effect was also observed for A. *glabripes* with the founder strain CAST/EiJ (Supplementary Fig. 4e). Additionally, we identified significant underrepresentation of *P. citreonigrum* in mice that were fed WES diet and in which the founder alleles were derived from the Cast/EiJ strain as opposed to mice in which the founder alleles were derived from the NZM2410/J strain (Supplementary Fig. 4f). In contrast, mice fed CAL diet and containing founder alleles derived from the Cast/EiJ strain showed significant overrepresentation of *P. citreonigrum* as opposed to mice on CAL diet in which the founder alleles were derived from the NZM2410/J mice.

## Validation and identification of new QTL associated with gut bacteria

After identifying that host genetics and its interaction with diet modulates fungi composition in the gut of AIL mice, we next investigated their effects on the composition of bacterial species. We identified QTL for both standing and active bacterial communities within the same cohort of AIL mice, again using the three models including Add, IntDiet, and IntSex (Figs. 1a and 4). In total, we identified 45 QTL for the standing bacterial communities and 38 QTL for the active bacterial communities (Supplementary Data 13). At the phylum level, we observed two QTL for the phylum *Proteobacteria* (RNA; IntDiet; Chr 9; LOD = 11.46) and *Actinobacteria* (DNA; Add; Chr 8; LOD = 6.76) harboring the genes *Gramd1b* and *Myom2*, respectively. Interestingly, *Gramd1b* is important for cholesterol homeostasis[87] and has been associated with 6-hydroxymethyl dihydropterin diphosphate biosynthesis in human gut microbiome GWAS in Dutch population[88]. At the class level, we identified QTL for *Clostridia* on Chr 9 (DNA; IntDiet; LOD = 10.51) and Chr 11 (RNA; Intsex; LOD = 10.1) containing the genes *Smad3* and *Tanc2*, respectively. Deficiency in *Smad3* protects against insulin resistance and obesity induced by a high-fat diet, while in gut it regulates TGF-β auto induction in *Clostridium butyricum*-activated dendritic cells[89]. Further, we identified an order level QTL for *Desulfovibrionales* (RNA; Add; LOD = 6.88), *Lactobacillales*, and *Bacteroidales* on Chr 11 (RNA; Add; LOD = 6.88), Chr 13 (DNA; InDiet, LOD = 10.6) and Chr 15 (DNA; Add, LOD = 6.78), respectively. At the family level, we identified 6 Add QTL for the families *Desulfovibrionaceae* (RNA; Chr 11; LOD = 7.01), *Planococcaceae* (DNA; Chr 12; LOD = 7.53), *Prevotellaceae* (DNA; Chr 3; LOD = 7.44; RNA; Chr 3; LOD = 7.42), *Peptostreptococcaceae* (genus *Romboutsia*, DNA; Chr 4; LOD = 7.31), and *Enterococcaceae* (genus *Enterococcus*, RNA; Chr 5; LOD = 7.18). The identified QTL for *Prevotellaceae* harbores the gene *Adgrl2* that encodes for a member of the latrophilin subfamily of G-protein coupled receptors, participating in the regulation of exocytosis. SNPs in *ADGRL2* have been previously found to be associated with gut microbiome GWAS in Japanese cohort[90] and genome-wide gut microbiota interaction with bone mineral density in the UK Biobank cohort[91].

For lower taxa ranks i.e., genus and species, overall, we identified 70 QTL (DNA: 39 QTL and RNA: 31 QTL) for Add (52 QTL), IntDiet (9 QTL), and IntSex (9 QTL) models (Supplementary Data 13). Of these, 59 QTL had been previously reported and 11 QTL are novel[92,93]. Out of the 70 QTL, 18 QTL mapped for OTUs assigned to *Clostridium*, 6 mapped for the *Roseburia* and its associated OTU, 5 mapped for *Marvinbryantia*, 4 mapped for *Ruminococcus*, and 4 for genus *Blautia*. Additionally, we mapped four species level OTUs (*C. populeti*, *O. valericigenes*, *R. hominis* and *M. formatexigens*) for both DNA and RNA QTL within the same intervals on Chr 1, Chr 14, Chr 15, and Chr 19. These 70 identified QTL include several known, but also so far unreported candidate genes. For example, the QTL controlling the genus *Marvinbryantia* and its species *M. formatexigens* (DNA and RNA) were mapped to Chr 19 harboring the candidate gene *Tcf7l2*. *Tcf7l2* encodes for transcription factor 7-like 2, a high mobility group (HMG) box-containing transcription factor that plays a role in the Wnt-signaling pathway and serves as a master regulator of insulin biosynthesis and

secretion[94]. Accordingly, the *TCF7L2* gene harbors risk SNPs with the strongest effect on T2D[95]. Interestingly, a higher abundance of the genus *Marvinbryantia* has been inversely correlated with insulin resistance[96]. We mapped *R. hominis* for both DNA and RNA on Chr 15 containing the candidate gene *Osmr*, which encodes for the Oncostatin-M specific receptor subunit beta. Increased levels of OSMR and its ligand OSM were identified in intestinal tissues from patients with IBD and closely correlated with disease severity on histopathological level[97]. In line with that decreased abundance of the butyrate-producing species *R. hominis* was associated with patients with ulcerative colitis.

Taken together, our results extend and support the role of host genetics and its interaction with diet in modulating the composition of intestinal standing and active bacterial communities.

### Common genetic control of fungi and bacteria in the mammalian gut

We identified a distinct role of diet in controlling the intestinal composition of the fungus and bacteria in the AIL mice. To investigate the role of host genetics on the co-regulation of fungi and bacteria in the gut, we next compared our mapped QTL to previously reported gut microbiome QTL[93,98,99]. We extracted 347 microbiome QTL from previous studies, which overlapped with 62% (28/45) of the bacterial standing communities and with 52% (20/38) of the bacterial active communities identified in the current study. Some of the bacterial QTL overlapped with previously identified QTL along the same taxonomical lineage (Supplementary Data 14). For example, DNA (IntDiet; Chr 9; LOD = 10.51) and RNA (Intsex; Chr11; LOD = 10.1) QTL for the class *Clostridia* overlapped with previously published QTL for the family *Lachnospiraceae* and the genus *Coprococcus* (both belonging to class *Clostridia*) on Chr 9, respectively[99]. The QTL for species level OTU for (DNA, LOD = 9.23; RNA, LOD = 7.35) *O. valericigenes* (order *Clostridiales*) overlapped with QTL for *Clostridiales* on chromosome 14[99]. Further, 60% of the identified fungal QTL (30/51) overlapped with previously reported gut microbiome QTL (Supplementary Data 14). For example, QTL for the genus *Penicillium* or different Basidiomycota species overlapped with QTL for species of the bacterial class *Clostridia* such as *Oscillospira* or *Coprococcus*. Notably, *Penicillium* was previously shown to negatively correlate with *Oscillospira* and *Clostridoides* infections[100]. Additionally, a decreased abundance of *Coprococcus* along with an increase in Basidiomycota was previously reported in IBD[32], suggesting that similar genetic elements regulate the composition of fungi and bacteria in the host via potentially common molecular pathways.

## Discussion

Fungi and bacteria colonize the mammalian gut forming a complex ecosystem consistent of dynamic microbe-microbe and host-microbe interactions that shape the immune system of the host. The role of diet has come to focus in studies investigating both bacterial and fungal associations with the host genetics, and its effects on the gut microbiome has been extensively studied in different organisms[101–105]. High-fat WES diet is one of the major causes for obesity seen in both mice and humans, which (in the mouse) correlated with high abundance of specific fungal genera such as *Aspergillus*[104] and *Candida*[106]. In the present study, we first analyzed the effects of diet on the composition of gut fungi in AIL mice. In accordance with previous study by Heisel et al.[104], examining the effects of high-fat diet (HFD) on the composition of gut fungi in C57BL/6 mice, no significant differences were found in fungi richness and evenness but rather in their relative composition in AIL mice fed WES diet. In both C57BL/6 and AIL mice, *Ascomycota* species dominated the gut ecosystem and were largely suppressed under high-fat and WES diets. Consistently, significant underrepresentation of members of the *Pleosporales* and the *Hypocreales* orders was observed in C57BL/6 and AIL mice. Thus, while AIL mice showed significant suppression in *Phoma, Ascochyta, Claviceps,* and

*Nectriaceae* taxa, C57BL/6 mice showed suppression of *Fusarium* and Didymellaceae following HFD. Remarkably, both WES and high-fat diets suppressed the relative abundance of the *Ascomycota* genus *Alternaria*, and induced overrepresentation of the *Basidomycota* genus *Wallemia* in the gut. Notably, we previously identified increased levels of *Wallemia* in a cohort of NZM2410/J mice fed Western diet and exhibiting lupus-nephritis, suggesting of a pro-inflammatory role of *Wallemia* species in the gut. In the AIL mice, the overrepresentation of the *Wallemia* genus was specifically attributed to the species *W. sebi*, a xerophilic fungus that is known for its remarkable genetic adaptations to osmotic stress[107]. In line with the suspected pro-inflammatory role of *Wallemia* sp., it was previously shown that overrepresentation of *W. sebi* in the murine gut is associated with exacerbated allergic airway disease and signs of intestinal inflammation[108], which are also exacerbated following exposure to HFD[109–111]. One of potential limitations of our study is the analysis of relative fungal abundance (standing communities) without profiling their activity (active communities). This was due to technical difficulties to achieve consistent quality and quantity of extracted fungal RNA from stool samples. Thus, future studies are required to profile the ITS2 region in mice at both the gene copy (DNA) and transcript (RNA) levels, which will reflect on the relative number and activity of the fungal taxa, respectively.

Fungal-bacterial interactions within the total microbial gut community influence fungal composition and specific traits in AIL mice. It has already been shown that bacteria and fungi can form "consortia" in which they coexist[56,112]. These interactions can play a role, especially during infections, when fungi like *Candida* are accompanied or challenged by bacteria or vice versa[113–116]. Within this study, comparing QTL for fungal and bacterial traits in AIL mice, no overlap between QTL of the two kingdoms was found. However, the identified correlations between fungal and bacterial taxa suggest the existence of putative key drivers within the gut ecosystem that are involved in the homeostasis regulation. Interestingly, four of the five identified key fungi genera, that were found to interact with most of the bacterial species in the gut, specifically *Claviceps, Alternaria, Davidella*, and *Wallemia*, were also significantly modulated by diet, suggesting that diet has a detrimental role on regulating microbe-microbe interactions in the gut, and is only one of the many potential mechanisms of how diet may modulate hosts' immune system.

Fungi are known to calibrate immunological responsiveness in humans that may improve the outcome of inflammatory disorders or infections. More recent reports have even implicated mycobiota in the pathogenesis of cancer in humans[117,118] and demonstrated that ablation of the commensal gut mycobiome by fluconazole treatment protected mice from developing pancreatic ductal adenocarcinoma[119]. Thus, dissecting the interplay between host genetics and diet on shaping the gut mycobiome composition for the first time in a mammal, may have profound implications on human health and disease. Here, we demonstrated that host genetics not only modulates the composition of bacteria but also of commensal fungi in the gut and explains the largest proportion (28/51 Add QTL) of the phenotypic variation in fungal composition. Interestingly, interaction of host genetics with diet reveals novel associations (IntDiet QTL) and explains ~33% (17/51 IntDiet QTL) of the phenotypic variation in fungi composition in the gut. The remaining 12% of the phenotypic variation, are attributed to sex (IntSex QTL) and are consistent with previous reports showing a shift in intestinal mycobiome composition in a gender-dependent manner[120]. These findings add to our understanding of how host genetics and diet shape the collective intestinal microbiome and warrant inclusion of diet and sex as covariates in human host–microbiome genome-wide association studies (mGWAS), as well as in experimental setups aiming at understanding the role of fungi in preclinical mouse models.

Using the AIL population that allows us to experimentally increase the actual recombination proportion across the genome and create a

genetically diverse pool, we identified QTL of an average size of $0.91 \pm 0.14$ Mb. This experimental design further allowed us to pinpoint several candidate genes within the identified QTL. Looking at the IntDiet model, we identified several candidate genes such as *Nos2*, *Vtn*, and *Ikbkb* that were previously reported to affect fungi composition in the host by potentiating phagocytic cell activation. In addition, we identified several interesting candidates such as *Taf4b*, *Tmc8*, *Wdr11*, and *Grk5* that were previously associated with different chronic metabolic and inflammatory diseases including T2D, cardiovascular disease, and/or IBD[69,70,74,76,77,80,81]. However, how this group of genes may modulate fungi composition in the host remains presently unknown.

To deepen our understanding on interference of diet with host genetics in shaping microbial traits including both fungi and bacteria, the evaluation of the identified candidate genes together with possible fecal transplantation studies in murine models will be of future interest. Large-scale dietary intervention studies could further unravel hidden correlations between host genetics and the different microbial kingdoms within the gut microbiome, as well as provide insights and potential new therapeutic approaches into common human disorders, such as obesity.

## Methods

### Animals and sample collection

All animal experiments were conducted in accordance with the European Community rules for animal care that were approved by the respective governmental administration of state of Schleswig-Holstein ("Ministerium für Energiewende, Landwirtschaft, Umwelt und ländliche Räume des Landes Schleswig-Holstein" in Kiel, Germany, reference number 27-2/13).

To study the interaction between host genetics and diet and its effect on the composition of microbiome in the gut, we performed QTL mapping. As the resolution of QTL mapping depends on the number of recombination events in the studied population, and on the effect size of the QTL, many thousands of such events in the genome are required. One approach to circumvent this difficulty is to use a population with high incidence of recombination, such as advanced intercross line (AIL). AIL is generated by intercrossing parental inbred lines for more than ten generations, resulting in chromosomes that are a mixture of the founder haplotypes. This approach provides improved mapping resolution, with minor allele frequencies and linkage disequilibrium (LD) comparable to values detected in isolated human populations[121]. In this study, we used the previously established four-way advanced intercross line (AIL)[26]. This line was generated by intercrossing four inbred parental strains MRL/MpJ, NZM2410/J, BXD2/TyJ, and Cast/EiJ (all purchased from Jackson Laboratory (Maine, USA) at equal strain and sex distribution for 20 generations with at least 50 breeding pairs per generation. Overall, we generated a cohort of 591 mice (Supplementary Data 1).

After weaning at 3–4 weeks, 591 offspring mice of either sex were transferred into separate cages and randomly allocated to one of the three different diets: control mouse chow (CON; #1320, Altromin Spezialfutter GmbH, Lage, Germany; $n_{CON} = 145$; $n_{males} = 54$; $n_{females} = 91$), caloric restriction (CAL; $n_{CAL} = 250$; $n_{males} = 111$; $n_{females} = 139$), and Western diet (WES; S0587-E020, ssniff Spezialdiäten GmbH, Soest, Germany; $n_{WES} = 196$; $n_{males} = 63$; $n_{females} = 133$). Control mouse chow (CON) was given *ad libitum*, while for caloric restriction (CAL), 60% of food amount (measured by weight) consumed by sex- and age-matched mice on control diet was given to mice once daily. For that the consumption of control food was consecutively measured in approx. 300 mice of 15th and 16th generation of this 4-way advanced intercross mouse strain. WES diet was given *ad libitum* and was rich in cholesterol, butter fat, and sugar. Animals were kept on the corresponding diet until the age of 6 months under specific pathogen-free conditions at 12 h light/dark cycle at the animal

facility of the University of Lübeck, Germany. At the age of 6 months, mice were sacrificed and cecum content samples were collected and processed for sequencing (Supplementary Data 1).

### Genotyping of AIL mice

Genomic DNA was isolated from the tail tips of AIL mice after six months using DNeasy Blood & Tissue Kit (Qiagen GmbH, Hilden, Germany) according to the manufacturer's instructions. Isolated DNA was quantified with the NanoDrop2.0 (Implen, Munich, Germany) and stored at −20 °C until further use. DNA was analyzed by MegaMUGA genotyping array (Neogen Genomics, Lincoln, Netherlands) covering 77,800 markers throughout the mouse genome.

### Bacterial DNA/RNA isolation and PCR

Bacterial DNA and RNA was isolated from murine cecum content samples that were stored at −20 °C in a RNAlater™ Stabilization Solution (Thermo Fisher Scientific, Waltham, MA, USA). Briefly, the cecum content was disrupted using the Speedmill Plus homogenizer (Analytik Jena, Jena, Germany) within Lysis Matrix E tubes (MPBio, Santa Ana, CA, USA) and DNA and RNA was isolated with the AllPrep DNA/RNA Mini Kit (Qiagen, Venlo, Netherlands) according to manufacturer's protocol. The isolation was carried out following the manufacturer's instructions with on-membrane DNA-digestion for 20 min during RNA isolation. Negative extraction controls (in total 23) were included for each RNA/DNA isolation process. Thirty ng of isolated RNA was transcribed into cDNA using the High Capacity cDNA Reverse Transcription Kit (Thermo Fisher Scientific, Waltham, MA, USA) following the manufacturer's instructions. Additionally, to control for possible contamination, a non-enzyme control was run during each transcription process. The isolated RNA and DNA were quantified using the Nanodrop 2000 spectrophotometer (Thermo Fisher Scientific, Dreieich, Germany) and checked for purity. Bacterial DNA and RNA-derived samples the V1−V2 regions of the bacterial 16S rRNA gene were amplified using dual indexing approach. All primers used in this study are indicated in the Supplementary Data 15. Briefly, primers contain broadly conserved bacterial primers 27F and 338R and P5 (forward) and P7 (reverse) sequences:

Forward 5′-AATGATACGGCGACCACCGAGATCTACAC XXXXXX XX <u>TATGGTAATTGT</u> AGAGTTTGATCCTGGCTCAG-3′ and

Reverse 5′-CAAGCAGAAGACGGCATACGAGAT XXXXXXXX <u>AGTC AGTCAGCC</u> TGCTGCCTCCCGTAGGAGT-3′).

In order to specifically tag PCR amplicon both primers contain unique eight-base multiplex identifier, designated as XXXXXXXX. To increase annealing temperature of sequencing primers, as recommended by Illumina, 12 nt long linker sequence (underlined) was added to primer sequences.

All PCR amplifications were conducted in a 20 μl volume containing 30 ng of either cDNA or DNA template using Phusion®Hot Start II DNA High-Fidelity DNA Polymerase (Thermo Fisher Scientific, Waltham, MA, USA). The cycling conditions were as follows: initial denaturation for 30 s at 98 °C; 20 cycles of 9 s at 98 °C, 30 s at 55 °C, and 30 s at 72 °C, final extension for 10 min at 72 °C.

### Fungal DNA isolation and PCR

Fungal DNA was isolated from murine cecum content samples, which were described above, using the DNeasy PowerLyzer PowerSoil Kit (Qiagen, Venlo, Netherlands) following the manufacturer's instructions with minor modifications. Briefly, samples were added to PowerBead tubes containing 750 μl PowerBead Solution, 60 μl C1 solution, and 20 μl Proteinase K (Qiagen, Venlo, Netherlands), and incubated in Eppendorf ThermoMixer® at 800 rpm at 50 °C for 2 hrs. After 2 h incubation, cecum contents were homogenized in a Precellys 24 tissue homogenizer (Bertin Technologies SAS, Montigny-le-Bretonneux, France). Subsequently, the nuclear ribosomal internal transcribed spacer 2 (ITS2) region was amplified using dual indexing approach.

Briefly, an internal transcribed spacer (ITS2) specific sequence primers fITS7 (forward) and ITS4 (reverse) were linked to a unique eight-base multiplex identifier (designated as XXXXXXXX) as described above for the 16S rRNA gene amplification. Primers also contain 10-nt pad sequence in order to prevent hairpin formation (underlined), and 2-nt linker sequence. Reverse primer was degenerated at one position. All primer sequences are described in Supplementary Data 15.

forward 5′- AATGATACGGCGACCACCGAGATCTACAC XXXXXX XX TATGGTAATT GG TCCTCCGCTTATTGATATGC-3′,

reverse 5′-CAAGCAGAAGACGGCATACGAGAT XXXXXXXX AGTC AGTCAG CC GTGA[AG]TCATCGAATCTTTG-3′

All PCR amplifications were conducted in a 25 µl volume using the Phusion® Polymerase (see above). The cycling conditions were as follows: initial denaturation for 30 s at 98 °C; 35 cycles of 9 s at 98 °C, 30 s at 50 °C, and 30 s at 72 °C, final extension for 10 min at 72 °C. Template-free reactions were performed with all forward and reverse primer combinations, for both bacterial and fungal PCR reactions, to exclude primer contamination.

## Library preparation and sequencing

All PCR products, both bacterial and fungal, were quantified on 1.5% agarose gel (Biozym, Hessisch-Oldendorf, Germany) that was run at 120 V for 5 min followed by 110 V for 1 h. Bacterial PCR products were quantified and directly mixed into equimolar subpools. The subpools were loaded on 1.5% agarose gel and further extracted with the GeneJet NGS Cleanup Kit (Thermo Fisher Scientific, Waltham, MA, USA). The fungal PCR products were extracted with the MinElute Gel Extraction Kit (Qiagen, Venlo, Netherlands). After extraction, concentration of each subpool was determined with the NEBNext Library Quantification Kit for Illumina (New England Biolabs, Frankfurt am Main, Germany) in accordance with the manufacturer´s instructions. The quantified subpools were combined into equimolar libraries, each library containing up to 300 samples for the bacterial V1–V2 region sequencing and separately for the fungal ITS2 region sequencing. Then, the libraries were purified using AMPure® Beads XP Kit (Beckman&Coulter, Brea, CA, USA) and quantified with the NEBNext Library Quantification Kit. Prior to sequencing, the average amplicon bp size of the library was determined by the Agilent Bioanalyzer with the Agilent High Sensitivity DNA Kit (Agilent, Santa Clara, CA, USA). Libraries were then sequenced on a MiSeq (Illumina, San Diego, CA, USA) using the MiSeq v3 2 × 300 cycles sequencing chemistry (Illumina, San Diego, CA, USA) at a 12pM (16S rRNA) or 17.5pM (ITS2) concentration together with 10% (16S rRNA) or 20% (ITS2) of PhiX control library (Illumina, San Diego, CA, USA).

## Data processing and statistical analysis

Raw bacterial and fungal data generated as gzipped FASTQ format files on the Illumina MiSeq sequencing platform consisted of 20 million reads for each library. The fungal ITS2 region data was analyzed using the open-source bioinformatic pipeline PIPITS (v2.7)[36]. Briefly, paired end reads were merged and reads below $q < 20$ were filtered out. Then, using "pipits_funits" reads belonging to ITS2 region were extracted. Next, we used UNITE dataset for reference-based chimera removal and UCHIME for de novo chimera removal by the vsearch algorithm (v.2.8) with $E = 0.5$.

Sequences were classified with the RDP Classifier 2.12[122] against the UNITE fungal database v8.2 (https://doi.org/10.15156/BIO/786369). After rarefaction, each sample was normalized to 5,000 sequences. Therefore, after quality control, 427 ITS2 samples were included for the downstream analyses. Reads derived from bacterial 16S rRNA sequencing were analyzed using QIIME1 (v1.9.1)[123] pipeline with python (v2.7.1) in anaconda environment. Briefly, unique barcode sequences were used to assign the pair-end reads to samples. Subsequently, pair-end reads merging and quality filtering was done with QIIME1. Then, demultiplexing and removal of tags and primers, as well as the detection and removal of chimera sequences was performed using

VSEARCH (v2.8)[48] followed by OTU clustering of sequences with ≥97% similarity. The OTUs were filtered with QIIME1 with a minimum number of 10.000 reads. After quality control, 557 16S rRNA DNA and 552 16S rRNA RNA samples were used for downstream analyses. Taxonomic assignments from domain down to genus level were performed against the RDP database using the Blast algorithm.

Subsequent, ecological analysis was done using the VEGAN (v2.5.7) library R package[124]. Alpha diversity for the composition of taxonomies and microbial and fungal communities was calculated using the Shannon metrics (mean). The non-Euclidean dissimilarity Bray Curtis, and the abundant Jaccard distance were used to compute beta diversity in 16S rRNA data and ITS data, respectively. To assess statistical significance for alpha and beta-diversity, we used generation and sex as covariates. To adjust covariates (sex and generation) for alpha diversity we first normalized Chao1, Shannon, and Simpson index using box-cox transformation (MASS v7.3.54R package). We then derived standardized residual by regressing (linear model, lm in R) the indices with covariates. Thereafter, these residuals were used for accessing statistical significance for dietary groups using one-way anova and post hoc test (Turkey procedure with fdr correction). Statistical significance in beta-diversity among the different dietary groups was determined using the distance-based redundancy analysis (dbRDA) in the VEGAN R package[124], which performs constrained (capscale) principle coordinate analysis. The "anova.cca" function (999 permutations) from the VEGAN R package was used to derive test statistics and P values while conditioning the model by both sex and generation. Since, post-hoc comparison are not available for constrained analysis we used "MANOVA.RM" (v0.5.2) R package for multivariate data analysis (based on resampling). MANOVA.RM R package assumes neither multivariate normality nor covariance homogeneity while performing multivariate data analysis. We used capscale scores (derived from constrained analysis) as response variable, diet as dependent variable with 1000 iterations for resampling ("MANOVA.wide") to derive multivariate statistical model. Afterwards, pairwise post hoc comparisons were performed using "simCI" function with Turkey's procedure to derive resampling based adjusted P values, estimates, and confidence intervals for every dietary comparisons. The current version of "simCI" do not provide exact P values below 0.01 therefore, we provide estimates and confidence intervals along with P values.

To identify, potential microbial and fungal biomarkers for traits (diet), we used the LEfSe algorithm (v1.1.2)[37], which combines standard statistical tests with biological consistency and effect relevance to determine the features (taxonomical ranks) that most likely explain the differences between classes (diet). For LEfSe we report only those taxa that were significant specifically for diet while excluding those which were also significant for either sex or generation. In addition to LEfSe we also performed indicator species analysis using "multipatt" function with default parameters in indicspecies R package(v1.7.9)[125] for all OTUs identified in bacterial and fungal community. For indicator species analysis diet sex and generation were analyzed as separate phenotypes. In current work, we only reported taxa that were significant specifically for diet while excluding those which were also significant for either sex or generation.

## QTL mapping

Using the plink toolset (v1.9)[126], non-informative SNPs were filtered out based on a minor allele frequency (maf) of >0.05, a missing geno probability of <0.1 and common homozygous SNPs among the founders resulting in 55,458 SNPs, which were used in downstream analysis. The Happy R package (v2.4)[127] was then used for probabilistic reconstruction of the AIL mouse genome in terms of that of the four founder strains. The posterior probability that each mouse was in one of the four possible genotype states was determined by using

a hidden Markov model at every adjacent marker interval across a chromosome. Based on posterior probability we calculated kinship matrix ("kinship.probs", DOQTL R package (v1.19)) that estimates intra-individual relationship[128]. For the phenotypic traits (microbiome and mycobiome), we first filtered out rare (abundance read count <5 present in <20% samples) taxonomical lineages (Phylum to genus and OTU). We used variance stabilizing transformation (DESeq2 R package, v1.12.3) for every lineage separately to normalize the count data matrix (abundance table)[129]. Thereafter, we calculated residual for each taxa using lmer R package while accounting for generation as fixed effect and cage as random effect. LOD scores for additive QTL were estimated by fitting residuals against posterior probabilities ($G$) with sex and diet as fixed effect and kinship matrix as random effect using scanone.eqtl (DOQTL R package)[130]. For interacting (diet and sex) QTL, we first estimated a full model where residuals were fitted against posterior probabilities ($G$) and product of these probabilities with interaction term ($G \times$ diet, sex) similar to additive QTL. For null model, we fitted the same model while excluding product of posterior probabilities with the interaction term. LOD scores for the interaction model were calculated as differences in the LOD score between full model and null model[131]. We estimated genome-wide significance ($P < 0.05$) by permutation procedure (1000 permutations). A 1.5 LOD drop described the confidence interval for a QTL.

We calculated the percentage of phenotypic variation explained separately for confounding, environmental, and genetic factors. Total variance explained by cage was calculated using 'VarCorr' from lme4 R package (v4.1) where cage was considered as random effect. The residuals phenotype obtained after regressing out the cage effect from the above model was fitted with linear model ('lm function in R') against sex, generation, and diet (fixed-effect). From this model, we derived the sum of squares for each variable and phenotypic variation was calculated as sum of square for each covariate divided by total sum of square. For calculating proportion for variation explained by QTL we used the equation $h^2 = 1 - 10^{-(2/n)\text{LOD}}$, where $h^2$ is the proportion of variation explained LOD is the LOD score for the QTL, and $n$ is the number of mice, as was previously described in the Package 'qtl' developed by Broman et al.[132].

To further analyze how specific fungal species are influenced by diet in a genetics-dependent manner, we assigned each mouse from the AIL to its founder strains using maximum posterior probability of the peak SNP. Assignment of AIL mice to the founder strains was performed independently for every genome-wide significant QTL. Afterwards, we performed association between the AIL mice derived founder strains with the standardized residuals (after regressing the abundances with cage, sex, and generation) of fungi species across individual diets (Supplementary Fig. 4).

### Correlation analysis

Correlation between mycobiome and microbiome at genus level within murine gut was investigated using FastSpar (v1.0) algorithm[57]. FastSpar algorithm is a python based implementation that infers interaction network for taxonomical units from count based compositional data. We applied FastSpar algorithm to identify correlated genus between bacteria and fungi. All $P$-values were corrected using the Benjamini–Hochberg multiple correction method. Afterwards, all the statistically significant correlated genus ($P < 0.05$) were used to infer bacteria-fungi network and visualized using Cytoscape (v3.8)[133] where key driver of the community (bacteria of fungi) were inferred based on number of correlating partners.

### Reporting summary

Further information on research design is available in the Nature Portfolio Reporting Summary linked to this article.

## Data availability

The raw sequencing data, i.e., FASTQ files for microbiome and mycobiome from AIL mice generated in this study have been deposited in the NCBI SRA under accession code PRJNA886734. Additionally, Plink formatted genotype data (bed, bim and fam files) for AIL mice is available at Zenodo platform (DOI: 10.5281/zenodo.7592055).

UNITE fungal database v8.2 (https://doi.org/10.15156/BIO/786369) was used. All other data supporting the findings of this study are provided in the Supplementary Information/Source Data file. Source data are provided with this paper.

## Code availability

All codes generated or used during the current study are available at Github repository (https://github.com/Yask-Gupta/QTL_MYCO) and at Zenodo platform (DOI: 10.5281/zenodo.7580792).

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

## Acknowledgements

Foremost we are grateful for the mentorship and continued support we had received from the late Detlef Zillikens. His inspiring enthusiasm and dedication to research will be deeply missed. This work has been financially supported by the Juniorförderung grant *"Characterization of host genetics, diet and microbiome interplay in systemic lupus erythematosus"* awarded by the University of Lübeck (YG), as well as Cluster of Excellence *Precision Medicine in Chronic Inflammation* (EXC 2167), the Collaborative Research Center *Pathomechanisms of Antibody-mediated Autoimmunity* (SFB 1526), Research Training Groups *Modulation of Autoimmunity* (GRK1727) (AV), *Genes Environment and Inflammation* (GRK1743) (AV), and the clinician scientist program (Deutsche Dermatologische Gesellschaft/Arbeitsgemeinschaft Dermatologische Forschung) (AV), all from the Deutsche Forschungsgemeinschaft and from the Schleswig-Holstein Excellence-Chair Program from the State of Schleswig Holstein.

## Author contributions

Y.G. performed the bioinformatics and statistical analyses. Y.G. and T.S. created the figures and tables. A.L.E. and F.B. performed NGS studies. T.S. supervised NGS studies. Y.G., A.V., R.J.L., and T.S. planned and set up the study. A.V. and K.B. conducted the animal experiments. Y.G., A.L.E., A.V., R.J.L., and T.S. wrote the manuscript. S.S.C., A.M.C., and C.D.S. critically revised the manuscript. Y.G., D.Z., and R.J.L. obtained funding for the work. All authors read, revised, and approved the final version of the manuscript.

## Competing interests

The authors declare no competing interests.
