## [Peer Review File · Nature Communications]

REVIEWER COMMENTS

Reviewer #1 (Remarks to the Author):

The mammalian gut contains rich diversity within its microbial ecosystem, which includes bacteria, archaea, protists, viruses, and fungi; dysregulation of these microbes have been associated with IBD and other diseases. It has been shown that host genetics and diet broadly influence the microbial community, but it is not entirely clear how host genetics and diet affect intestinal fungi communities. Overall, the data analysis and interpretation in this study, as well as the methodology, are sound, and with some adjustments, the manuscript could be improved:

1. It would be helpful to have greater understanding of the rationale behind using AIL mice. For instance, do the parent strains, and their alleles, offer a strategic advantage in this study?
2. The largest difference in N exists between the mice receiving the control diet (N=145) and calorie-reduced mice (N=250). Thus, does the group receiving the control diet have an N too low for adequate mapping power, especially regarding the interactive effects of diet and genotype?
3. More details on the calorie restriction diet are needed.
4. The phylum and genus data (Fig. 1B) show considerable diversity within each dietary group. It would be easier to contextualize these data if the expected values from B6 mice (if available) were discussed somewhere in the manuscript, even the Discussion.
5. A broader issue: the Discussion could be improved by expanding on how this study improves upon existing knowledge on the topic. After all, many of the QTL found in this study overlap with QTL discovered in past studies, and candidate genes from past studies were subsequently confirmed.

Reviewer #2 (Remarks to the Author):

The study conducted by Yask Gupta et al investigated the effects of diet and host genetics on the murine intestinal mycobiome. For a long time, the community is interested in learning the impact of genetics on the gut mycobiome. This study is novel and interesting in this regard. However, I do have below concerns:

1. The study is broad and somewhat loses focus in the description of the results. While the effects of host genetics and diet on gut mycobiome are the novel parts of this study, the authors should focus

more on these parts to describe their findings. For some parts, such as the effect of diets and host genetics on the gut bacterial microbiome are well-reported by others. I believe the authors used a lot of strength to describe their findings on these parts.

2. The title says “the impact of host genetic and diet on...”, while the manuscript starts with diet and ends with host genetics, the sequence needs to be refined.

3. The study investigated the bacterial microbiome at the DNA and RNA levels, representing the standing and active communities respectively. As the novel part of the study, I wonder why the fungi (mycobiome) were not studied in the standing and active communities respectively.

4. Lines 248-256, when differentiating the effects of different factors, it's better to state which method was used to quantify and differentiate the effect sizes.

5. Lines 314-318, CARD9 is critical for sensing and controlling fungi, the authors did not specify how this gene stands out in their analysis. The authors should also state the correlation direction and coefficient.

6. The authors should do additional analysis to study if (or how) diet and genetics are intertwined to influence gut fungi. For example, if there are specific fungal species that are influenced by diet in a genetics-dependent manner, or vice versa

7. Line 76 drive should be driver

We thank the editors and reviewers for carefully reading our work and for their thoughtful comments, which allowed us to improve and extend our manuscript.

Below is a detailed point-by-point response to all the comments

Reviewers' comments:

Reviewer 1 (Remarks to the Author):

The mammalian gut contains rich diversity within its microbial ecosystem, which includes bacteria, archaea, protists, viruses, and fungi; dysregulation of these microbes have been associated with IBD and other diseases. It has been shown that host genetics and diet broadly influence the microbial community, but it is not entirely clear how host genetics and diet affect intestinal fungi communities. Overall, the data analysis and interpretation in this study, as well as the methodology, are sound, and with some adjustments, the manuscript could be improved:

1. It would be helpful to have greater understanding of the rationale behind using AIL mice. For instance, do the parent strains, and their alleles, offer a strategic advantage in this study?

We thank Reviewer 1 for this question. Genetic investigation in model organisms offers some advantages over human genome-wide association studies, for example controlled environmental influence and possibility of controlled genetic diversity. However, the use of classical F2 crosses for QTL mapping is limited by low resolution, due to insufficient recombination events, so that QTL can be assigned only to a large haplotype block¹. Commercial outbred mouse lines have higher recombination rate and can provide improved mapping resolution, with minor allele frequencies and linkage disequilibrium (LD) comparable to values detected in isolated human populations². However, in some cases, when a given causal allele is rare, use of commercial outbred mouse lines can lead to loss of statistical power³. Advanced intercross lines (AIL) are created by intercrossing of inbred mouse strains beyond F2 generation⁴. Each additional generation leads to further recombination events and reduction of LD between adjacent markers, allowing more precise mapping⁵. As each inbred strain contributes equally to AIL, all genetic variants are common and all polymorphic markers are informative, allowing identification, from which inbred strain the genetic region is inherited^{3,5}. The use of AIL strain in our project allows to avoid loss of statistical power from rare alleles and keep high mapping resolution even with relatively small mouse cohort.

We added the following paragraph to the Methods section (Please refer to the “Animals and sample collection” paragraph in our manuscript, lines: 552-567): “To study the interaction between host genetics and diet and its effect on the composition of microbiome in the gut, we performed QTL mapping. As the resolution of QTL mapping depends on the number of recombination events in the studied population, and on the effect size of the QTL, many thousands of such events in the genome are required. One approach to circumvent this difficulty is to use a population with high incidence of

recombinants, such as advanced intercross line (AIL). AIL is generated by intercrossing parental inbred lines for more than ten generations, resulting in chromosomes that are a mixture of the founder haplotypes. This approach provides improved mapping resolution, with minor allele frequencies and linkage disequilibrium (LD) comparable to values detected in isolated human populations². In this study, we used the previously established four-way advanced intercross line (AIL)⁶. This line was generated by inter-crossing four inbred parental strains MRL/MpJ, NZM2410/J, BxD2/TyJ, and Cast/EiJ at equal strain and gender distribution for 20 generations with at least 50 breeding pairs per generation. Overall we generated a cohort of 591 mice (Supplementary Table 1)”.

2. The largest difference in N exists between the mice receiving the control diet (N=145) and calorie-reduced mice (N=250). Thus, does the group receiving the control diet have an N too low for adequate mapping power, especially regarding the interactive effects of diet and genotype?

We thank Reviewer 1 for this important comment. Indeed, in our study we had a smaller cohort size of mice receiving control diet (N=145) as opposed to mice receiving calorie-reduced diet (N=250) or mice fed western diet (N=196). Though low sample size can reduce the power to detect QTL for rare and low abundant species, in our QTL analysis we focussed on highly abundant taxonomical groups. This is in line with previous studies where small group of genetically distinct mice had sufficient power to detect QTL for highly abundant microbial taxa^{7,8}. For example, Snijders *et al.* identified 169 joint QTL intervals for ten or more OTUs using 30 collaborative cross (CC) strains and McKnite *et al.* identified 5 QTL using 30 BXD strains, which resulted from the combination of C57BL/6J and DBA/2J genomes. In comparison in our study, we used 145 genetically distinct strains in the control group only and 591 strains across all three dietary groups to identify additive QTL and IntDiet QTL. In addition, to control for unequal sample size distribution between the three dietary groups in our study, which can lead to false positive QTL we performed a permutation test. We used 1000 permutations to identify the maximum LOD score that can occur by random chance for every phenotypic trait (across all microbiome taxa). To identify Gene x Diet QTL (among other QTL), the permutations were performed by shuffling the phenotypic trait across the three groups of mice while keeping the diet and the genotype variables constant. Using this data, we calculated the genome-wide maximum LOD score, which represents the highest LOD score generated by random chance and used it to set a threshold of statistical significance (genome-wide significance threshold <0.05). We only reported LOD scores, which were higher from that established threshold. Using this approach, we identified several Gene x Diet QTL that showed significant difference in the distribution of several fungi species across the three dietary groups (Supplementary Fig. 4).

3. More details on the calorie restriction diet are needed.

We thank the Reviewer for this comment. In our study, control mouse chow (CON) was given *ad libitum*, while caloric restriction (CAL) was performed by reducing 40% of the control mouse chow consumed by the age- and sex- matched controls. For that the consumption of control food was consecutively measured in approx. 300 mice of 15th and

16th generation of this 4-way advanced intercross mouse strain. For caloric restriction, 60% of food amount (measured by weight) consumed by sex- and age-matched mice on control diet was given to mice once daily. WES diet was given *ad libitum* and was rich in cholesterol, butter fat, and sugar. This information was added to the Methods section. Please refer to the “Animals and sample collection” paragraph (lines 572-579) in our manuscript.

4. The phylum and genus data (Fig. 1B) show considerable diversity within each dietary group. It would be easier to contextualize these data if the expected values from B6 mice (if available) were discussed somewhere in the manuscript, even the Discussion.

We thank Reviewer 1 for this important question. As per Reviewer’s 1 suggestion, we compared our findings to previously published data examining the effects of high fat diet (HFD) on the composition of intestinal mycobiome in C57BL/6 mice by Heisel *et al.*⁹

We added the following paragraph to the Discussion (lines 473-493): “In accordance with previous study by Heisel *et al.*⁹, examining the effects of high-fat diet (HFD) on the composition of gut fungi in C57BL/6 mice, no significant differences were found in fungi richness and evenness but rather in their relative composition in AIL mice fed WES diet. In both C57BL/6 and AIL mice, Ascomycota species dominated the gut ecosystem and were largely suppressed under high-fat and WES diets. Consistently, significant underrepresentation of members of the Pleosporales and the Hypocreales orders was observed in C57BL/6 and AIL mice. Thus, while AIL mice showed significant suppression in *Phoma*, *Ascochyta*, *Claviceps* and Nectriaceae taxa, C57BL/6 mice showed suppression of *Fusarium* and Didymellaceae following HFD. Remarkably, both WES and high-fat diets suppressed the relative abundance of the Ascomycota genus *Alternaria*, and induced overrepresentation of the Basidiomycota genus *Wallemia* in the gut. Notably, we previously identified increased levels of *Wallemia* in a cohort of NZM2410/J mice fed Western diet and exhibiting lupus-nephritis, suggesting of a pro-inflammatory role of *Wallemia* species in the gut. In the AIL mice, the overrepresentation of the *Wallemia* genus was specifically attributed to the species *W. sebi*, a xerophilic fungus that is known for its remarkable genetic adaptations to osmotic stress¹⁰. In line with the suspected pro-inflammatory role of *Wallemia* sp., it was previously shown that overrepresentation of *W. sebi* in the murine gut is associated with exacerbated allergic airway disease and signs of intestinal inflammation¹¹, which are also exacerbated following exposure to HFD¹²⁻¹⁴”.

5. A broader issue: the Discussion could be improved by expanding on how this study improves upon existing knowledge on the topic. After all, many of the QTL found in this study overlap with QTL discovered in past studies, and candidate genes from past studies were subsequently confirmed.

We thank Reviewer 1 for raising this important comment, which allowed us to further improve and extend our discussion to highlight the contribution of this study to the field. Please refer to the Discussion section in the revised version of our manuscript (lines 465-550).

Reviewer 2 (Remarks to the Author):

The study conducted by Yask Gupta et al investigated the effects of diet and host genetics on the murine intestinal mycobiome. For a long time, the community is interested in learning the impact of genetics on the gut mycobiome. This study is novel and interesting in this regard. However, I do have below concerns:

1. The study is broad and somewhat loses focus in the description of the results. While the effects of host genetics and diet on gut mycobiome are the novel parts of this study, the authors should focus more on these parts to describe their findings. For some parts, such as the effect of diets and host genetics on the gut bacterial microbiome are well-reported by others. I believe the authors used a lot of strength to describe their findings on these parts.

We thank Reviewer 2 for this comment. We agree with the Reviewer that the effects of host genetics and its interaction with diet on fungi composition in the gut are the novel parts of this study. However, based on the previously established fungal-bacterial interactions within the mammalian gut ecosystem, their modulation by diet, as well as the previously reported role of host genetics and diet on controlling bacterial composition in the gut, we think it is important to discuss the fungi and bacteria composition and their interaction in our work. As per suggestion of the Reviewer, to try to streamline our narrative and improve the sequential presentation of our results, we included the paragraph “Co-regulation and inter-domain correlations of fungi and bacteria in the gut ecosystem” right after the description of bacterial composition in the gut. Please refer to the Results section lines 236-279 in our manuscript.

2. The title says “the impact of host genetic and diet on...”, while the manuscript starts with diet and ends with host genetics, the sequence needs to be refined.

We agree with Reviewer 2. To better reflect the sequence of our findings we modified the title to: “Impact of diet and host genetics on the murine intestinal mycobiome”.

3. The study investigated the bacterial microbiome at the DNA and RNA levels, representing the standing and active communities respectively. As the novel part of the study, I wonder why the fungi (mycobiome) were not studied in the standing and active communities respectively.

We thank Reviewer 2 for raising this important point. Study of the active fungal communities requires extraction of good quality RNA from stool biomass. Fungal cell walls are known to be rigid structures with high amount of polysaccharides. Rigidity of fungal cell wall currently does not allow the use of standard RNA extraction protocols established for animal cells or bacteria and requires additional aggressive mechanical, chemical or enzymatic treatment. As composition of fungal cell wall can considerably vary

between genera, it often requires adjustment of lysis protocol for different genera, that complicates the extraction of RNA from complex biosamples such as stool biomass. The high molecular weight polysaccharides of fungal cell wall can hinder extraction of RNA and reduce its quality and yield^{15,16}. Further, phenolic compounds produced by many fungi can lead to RNA precipitation and inhibit further downstream applications¹⁷. In our experiments, we were, however, were also interested in extracting fungal RNA from our samples. However, using several protocol variations, we were not able to achieve consistent results of extracted RNA, so that we did not proceed with studying active fungal communities.

To address this valid point, we added the following paragraph to the Discussion section as part of the limitations of our study (lines 493-499): “One of potential limitations of our study is the analysis of relative fungal abundance (standing communities) without profiling their activity (active communities). This was due to technical difficulties to achieve consistent quality and quantity of extracted fungal RNA from stool samples. Thus, future studies are required to profile the ITS2 region in mice at both the gene copy (DNA) and transcript (RNA) levels, which will reflect on the relative number and activity of the fungal taxa, respectively.”

4. Lines 248-256, when differentiating the effects of different factors, it's better to state which method was used to quantify and differentiate the effect sizes.

Following your suggestion, we added the following paragraph to the Methods section (please refer to the paragraph “QTL mapping” lines 721-731): “We calculated the percentage of phenotypic variation explained separately for confounding, environmental, and genetic factors. Total variance explained by cage was calculated using ‘VarCorr’ from lme4 R package where cage was considered as random effect. The residuals phenotype obtained after regressing out the cage effect from the above model was fitted with linear model (‘lm function in R’) against sex, generation and diet (fixed-effect). From this model, we derived the sum of squares for each variable and phenotypic variation was calculated as sum of square for each covariate divided by total sum of square. For calculating proportion for variation explained by QTL we used the equation $h^2 = 1 - 10^{-(2/n)LOD}$, where h^2 is the proportion of variation explained LOD is the LOD score for the QTL, and n is the number of mice, as was previously described in the Package ‘qtl’ developed by Broman *et al.*¹⁸ “

5. Lines 314-318, CARD9 is critical for sensing and controlling fungi, the authors did not specify how this gene stands out in their analysis. The authors should also state the correlation direction and coefficient.

We thank Reviewer 2 for raising this comment and apologize for this omission and not precise wording. We did not identify the *Card9* gene in our analysis, but rather the candidate gene *lkbkb* in the IntDiet QTL (Chr 8, LOD=8.77) for *Malassezia restricta*. *M. restricta* was recently shown to promote the production of NF-kB-induced pro-inflammatory cytokines in myeloid phagocytes in the gut of patients with Crohn’s disease in a CARD9-dependent manner¹⁷. Thus, in our study, we mentioned the association

between *Ikbkb* and *Card9*, since CARD9 serves as an adaptor protein for c-type lectin receptors, which recognize fungi and activate the NF-Kb signalling pathway. The *Ikbkb* gene is coding for the inhibitor of nuclear factor kappa-B (NF-kB) kinase subunit beta (IKBKB), a key molecule in the NF-kB signalling pathway. IKBKB phosphorylates Ikb molecule, which interacts with the NF-kB transcription factors (RelA and p50) and renders them inactive. Once phosphorylated, the Ikb molecule is degraded followed by the release of NF-kB transcription factors and activation of the NF-Kb pathway in the nuclei. Thus, various signalling pathways that activate the NF-Kb transcription factors converge at the level of IKBKB. The gene *Ikbkb* is one of the candidate genes identified within the *M. restricta* QTL and it stands out in the analysis due to its previous association with *M. restricta*.

Following Reviewer's comment, we included a more detailed description in regards the association between *Ikbkb* and the *M. restricta* QTL (please refer to results section paragraph, lines: 358-371). Additionally, following this comment, we modified Table 1 and Supplementary Table 13 to include the percentage of the phenotypic variation for each QTL explained by sex, genetics, generation, diet, and cage indicated as %h² Sex, %h² LOD, %h² Gen, %h² Diet, and % h² Cage, respectively. Additionally, we also calculated the spearman's correlation coefficient for each QTL for sex, diet, and generation which are indicated as $\rho(\text{Sex})$, $\rho(\text{Diet})$ and $\rho(\text{Gen})$. The correlation direction is indicated by (+) or (-) signs describing a positive or negative correlation, respectively. Please refer to updated Table 1 and Supplementary Table 13.

6. The authors should do additional analysis to study if (or how) diet and genetics are intertwined to influence gut fungi. For example, if there are specific fungal species that are influenced by diet in a genetics-dependent manner, or vice versa

Thank you for raising this important point. To analyse how specific fungal species are influenced by diet in a genetics-dependent manner, we assigned each mouse from the AIL to its founder strains using maximum posterior probability of the peak SNP. Assignment of AIL mice to the founder strains was performed independently for every genome-wide significant QTL. Afterwards, we performed association between the AIL mice derived founder strains with the standardized residuals (after regressing the abundances with cage, sex, and generation) of fungi species across individual diets (Supplementary Fig. 4).

Across the six-studied fungal species QTL, we identified several examples illustrating gene by diet interactions (GenexDiet) that modulate the composition of distinct fungal species. For example, founder alleles at the locus (Chr 1: 133,222,151-133,859,717 BP) were associated with the distribution of *P. citreonigrum*. We identified significant underrepresentation of *P. citreonigrum* in the mice where founder allele was derived from Cast/EiJ mice, which were exposed to western diet as opposed to mice where founder allele from derived from the NZM2410/J on western diet. Conversely, mice associated with founder allele of Cast/EiJ mice fed calorie- reduced diet, show significant overrepresentation of *P. citreonigrum* as opposed to mice where founder alleles were associated with NZM2410/J mice, which showed underrepresentation of *P. citreonigrum* on similar diet.

We summarized our findings in Supplementary Fig. 4 and added a paragraph to the results section in the manuscript (lines 372-384): “Diet dependent QTL (IntDiet) were observed for five species *Penicillium citreonigrum*, *M. restricta*, *Penicillium spathulatum*, *A. nidulans*, and *Aspergillus glabripes*. Deconvolving AIL mice to the founder strains (see Methods section) showed higher abundance of *A. nidulans*, *M. restricta*, and *P. spathulatum* in AIL mice containing MRL/MpJ allele under CAL and CON diets in comparison to WES diet (Supplementary Fig. 4a-d). Same effect was also observed for *A. glabripes* containing alleles associated derived from founder strain CAST/EiJ (Supplementary Fig. 4e). Additionally, we identified significant underrepresentation of *P. citreonigrum* in mice that were fed WES diet and in which the founder alleles were derived from the Cast/EiJ strain as opposed to mice in which the founder alleles were derived from the NZM2410/J strain (Supplementary Fig. 4f). In contrast, mice fed CAL diet and containing founder alleles derived from the Cast/EiJ strain showed significant overrepresentation of *P. citreonigrum* as opposed to mice on CAL diet in which the founder alleles were derived from the NZM2410/J strain.

7. Line 76 drive should be driver

We thank Reviewer 2 for careful reading of the manuscript. We corrected line 76.

REFERENCES:

1. Moradi Marjaneh, M. *et al.* QTL mapping of complex binary traits in an advanced intercross line. *Anim Genet* **43 Suppl 1**, 97–101 (2012).
2. Nicod, J. *et al.* Genome-wide association of multiple complex traits in outbred mice by ultra-low-coverage sequencing. *Nat Genet* **48**, 912–918 (2016).
3. Gonzales, N. M. *et al.* Genome wide association analysis in a mouse advanced intercross line. *Nat Commun* **9**, 5162 (2018).
4. Darvasi, A. & Soller, M. Advanced intercross lines, an experimental population for fine genetic mapping. *Genetics* **141**, 1199–1207 (1995).
5. Parker, C. C. & Palmer, A. A. Dark matter: are mice the solution to missing heritability? *Front Genet* **2**, 32 (2011).
6. Vorobyev, A. Gene-diet interactions associated with complex trait variation in an advanced intercross outbred mouse line. *Nature communications* **10**, 4097, 10 1038 41467-019-11952- (2019).
7. McKnite, A. M. *et al.* Murine gut microbiota is defined by host genetics and modulates variation of metabolic traits. *PLoS One* **7**, e39191 (2012).
8. Snijders, A. M. *et al.* Influence of early life exposure, host genetics and diet on the mouse gut microbiome and metabolome. *Nat Microbiol* **2**, 16221 (2016).
9. Heisel, T. *et al.* High-Fat Diet Changes Fungal Microbiomes and Interkingdom Relationships in the Murine Gut. *mSphere* **2**, e00351-17 (2017).
10. Padamsee, M. *et al.* The genome of the xerotolerant mold *Wallemia sebi* reveals adaptations to osmotic stress and suggests cryptic sexual reproduction. *Fungal Genet Biol* **49**, 217–226 (2012).

11. Wheeler, M. L. *et al.* Immunological Consequences of Intestinal Fungal Dysbiosis. *Cell Host Microbe* **19**, 865–873 (2016).
12. Cheng, L. *et al.* High fat diet exacerbates dextran sulfate sodium induced colitis through disturbing mucosal dendritic cell homeostasis. *Int Immunopharmacol* **40**, 1–10 (2016).
13. Wood, L. G., Garg, M. L. & Gibson, P. G. A high-fat challenge increases airway inflammation and impairs bronchodilator recovery in asthma. *J Allergy Clin Immunol* **127**, 1133–1140 (2011).
14. Allegra, C. J., Egan, G. F., Drake, J. C., Steinberg, S. M. & Swain, S. M. The treatment of metastatic breast cancer with 5-fluorouracil and leucovorin. *Adv Exp Med Biol* **244**, 107–112 (1988).
15. Sánchez-Rodríguez, A. *et al.* An efficient method for the extraction of high-quality fungal total RNA to study the *Mycosphaerella fijiensis*-*Musa* spp. Interaction. *Mol Biotechnol* **40**, 299–305 (2008).
16. Shu, C., Sun, S., Chen, J., Chen, J. & Zhou, E. Comparison of different methods for total RNA extraction from sclerotia of *Rhizoctonia solani*. *Electronic Journal of Biotechnology* **17**, 50–54 (2014).
17. Leite, G. M., Magan, N. & Medina, Á. Comparison of different bead-beating RNA extraction strategies: an optimized method for filamentous fungi. *J Microbiol Methods* **88**, 413–418 (2012).
18. Broman, K. W., Wu, H., Sen, S. & Churchill, G. A. R/qtl: QTL mapping in experimental crosses. *Bioinformatics* **19**, 889–890 (2003).

REVIEWERS' COMMENTS

Reviewer #1 (Remarks to the Author):

All issues have been adequately addressed.

Reviewer #2 (Remarks to the Author):

I thank the authors' efforts to address all of my concerns which are satisfied in this version.